# DEFECT TRANSFER GAN: DIVERSE DEFECT SYNTHESIS FOR DATA AUGMENTATION

## ABSTRACT

Large amounts of data are a common requirement for many deep learning approaches. However, data is not always equally available at large scale for all classes. For example, on highly optimized production lines, defective samples are hardly acquired while non-defective samples come almost for free. The defects however often seem to resemble each other, e.g., scratches on different products may only differ in few characteristics. In this work, we propose to make use of the shared characteristics by transferring a stylized defect-specific content from one type of background product to another. Moreover, the stochastic variations of the shared characteristics are captured, which also allows generating novel defects from random noise. These synthetic defective samples enlarge the dataset and increase the diversity of defects on the target product. Experiments demonstrate that our model is able to disentangle the defect-specific content from the background of an image without pixel-level labels. We present convincing results on images from real industrial production lines. Also, we show consistent gains of using our method to enlarge training sets in classification tasks.

## 1 INTRODUCTION

Automated Visual Inspection (AVI) is vital for quality control in modern production lines. Despite the fact that AVI has been studied for decades, it remains a challenging task with many open research questions await to be answered. One of the main challenges in data-driven AVI is the acquisition of suitable training data. This is for two reasons: First, collecting a vast amount of labelled data is usually labor-intensive and time-consuming. In many cases, even experts are required to identify where and what to look for. However, the acquired label information is task-specific and cannot be reused or transferred to a new task in most cases. Thus, the tedious labelling process must be repeated for each new product, even if its defect is similar to other products in people's eyes. Second, in real-world scenarios such as highly optimized production lines, a more severe problem emerges: data imbalance. Only very few defective parts are produced by design. Moreover, the acquired anomaly images from a single product are lacking diversity and may not capture the full defect distribution. Training a robust deep neural network model in such conditions is very challenging.

Since collecting sufficient real-world defective samples is impractical, algorithms to synthesize required images became a focus in research. Image synthesis through Generative Adversarial Networks (GANs) (Goodfellow et al., 2014) has shown promising performance in recent years. But it also requires large amounts of balanced data which are not available in most industrial use cases, in particular for irregular defect patterns and large variation. Therefore, GANs tend to overfit to the training examples when trained with little data (Karras et al., 2020a).

In this work, we tackle these issues by exploiting cross-domain information: we first define two sets of *domains*—foreground domains and background domains. The foreground domain describes a set of images that contains a specific foreground content to be grouped into a distinctive category, and each content has a different *style*. The background domain instead is considered as a group of images that shares similar structural appearance over the whole image. For example, we can set foreground domains as defect types and background domains as product types while the styles of defects indicate their artistic looks such as light or heavy strokes. Building upon StarGAN v2 (Choi et al., 2020), the concept underlying this work is to transfer and generate foreground contents with a variety of styles across different background domains, as illustrated in Figure 1.

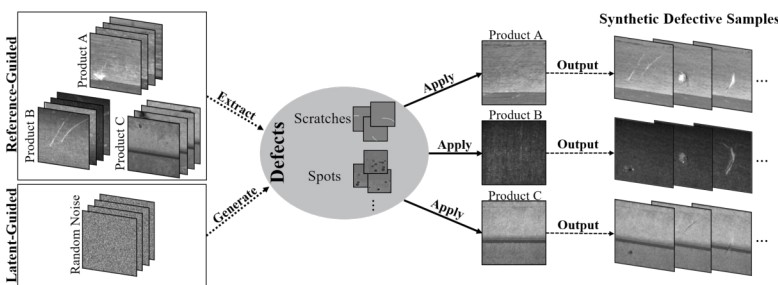

Figure 1: The underlying concept of DT-GAN is to transfer and generate foreground contents with a variety of artistic styles (e.g., light / heavy strokes) across different background domains.

The contributions of this work are three-fold: First, we introduce Defect Transfer GAN (DT-GAN), a model that learns transferring existing foreground content and generating novel contents onto different backgrounds at the same time. In the real-world scenario, it allows defect inspection networks to learn from a variety of synthetic defective images by composing the foreground defects together with various non-defective images from different products. Second, DT-GAN is able to disentangle the foreground defect-specific content and the defect-irrelevant background in a weakly-supervised manner. Third, extensive experiments show that our method can generate diverse and real-looking defective samples even for products with only 20 real defective images. These defective images generated by DT-GAN boost the performance in defect inspection networks significantly.

## 2  RELATED WORK

GANs have shown their power in many computer vision tasks such as image synthesis (Lučić et al., 2019), style translation (Johnson et al., 2016), super-resolution (Ledig et al., 2017), image impainting (Pathak et al., 2016) and many other applications. To quantify the performance of GANs, visual quality and the diversity of generated images are considered as two of the most important criteria. Recent models address these requirements either by dedicated loss functions (Mao et al., 2019b; Yang et al., 2019) or architectural design (Brock et al., 2019). StyleGAN v2 (Karras et al., 2020b), the latest state-of-the-art model in image synthesis, introduces stochastic variation in image generating process by adding per-pixel noise after each convolution. However, it is non-trivial to adapt the model to transform given input images due to the design of the generator.

In contrast, image-to-image translation methods (Isola et al., 2017) provide a way to recover the connection between inputs and the generated images while encouraging diversity. For example, Zhu et al. (2017b) and Huang et al. (2018) impose consistent mappings in latent space to achieve the goal. Some approaches (Ma et al., 2019; Park et al., 2019) use reference images as guidance to generate diverse outputs. Mokady et al. (2020) further extends the translation task from styles to contents. It learns to identify a specific content in a given input (e.g., a specific pair of glasses) and transfer it to the target image. However, aforementioned methods only consider the translation between two domains and their extension to multiple domains is non-trivial.

Surface defect detection is one of the important tasks in real-world industrial manufacturing. It aims at identifying and classifying defects with the help of machine vision. Traditional methods (Ngan et al., 2011) build models upon hand-crafted feature extractors, which are unstable and outperformed by deep learning based models. However, the performance and generalization ability of deep learning approaches are restricted due to limited number of defective samples in real-world scenarios. Data augmentation aims to enrich the training dataset by introducing different kinds of invariance for the model to capture. Several recent works (Niu et al., 2020; Zhang et al., 2021) have proposed to adopt GANs as a data augmentation method to generate realistic defective samples. Among them, Defect-GAN (Zhang et al., 2021) tries to capture the stochastic variation within defects by mimicking the defacement and restoration processes. However, it still learns a deterministic mapping between inputs and outputs while DT-GAN achieves multi-modality by varying styles. Moreover, our method can generate realistic defects with sophisticated patterns copied from real-world defective samples.

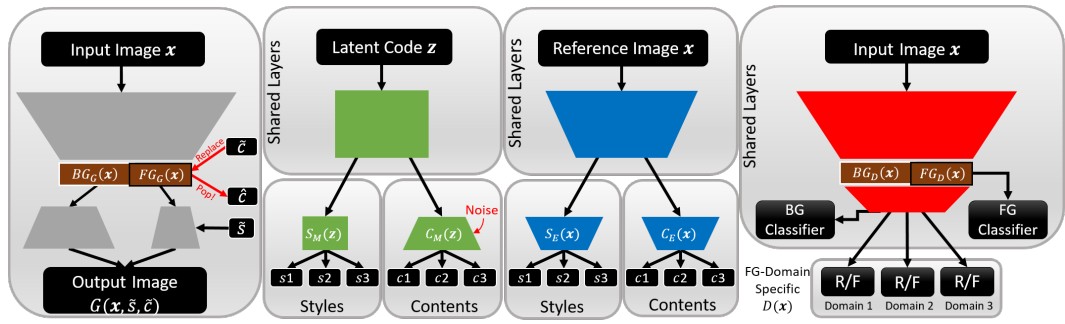

Figure 2: Overview of all modules in DT-GAN.

## 3 METHODOLOGY

Our primary aim is to perform unpaired image-to-image translation across multiple foreground domains within a single model. In our use case, the foreground domains refer to the defect types, which means we want to achieve translations between different types of defects while the background remains unaffected. We assume that there is always an adequate amount of normal samples (e.g., non-defective) available, while anomaly samples are rare and hard to acquire.

### 3.1 PROPOSED FRAMEWORK

Our framework builds on StarGAN v2, a multimodal image-to-image translation model. Given an input image $\mathbf{x} \in \mathcal{X}$ and an arbitrary domain $y \in \mathcal{Y}$, StarGAN v2 generates a domain specific style code in a learned style space and outputs an image that is stylized to fit the domain of $y$. Its network architecture consists of four modules: a generator, a mapping network, a style encoder and a discriminator. We modify and extend all four modules (see Figure 2) and describe the key differences in details as below.

**Style-Content Separation.** Given a latent code $\mathbf{z}$ and a domain $y$, the mapping network $M$ (Figure 2(b)) generates a style code $\mathbf{s} = M_y(\mathbf{z})$ and a domain specific content $\mathbf{c} = M_y(\mathbf{z})$ in different branches. It is worth mentioning that $M_y$ here denotes an output of $M$ corresponding to the domain $y$. This feature allows our method to separately model the structural appearance (i.e. content) and its artistic looks (i.e. style), which is essential because applying different styles to the same content enriches the diversity of outputs. By randomly sampling $\mathbf{z}$ from a standard normal distribution and $y$ from all available foreground domains, $M$ is able to produce diverse style codes and domain specific contents.

The encoder $E$ (Figure 2(c)) extracts the style code $\mathbf{s} = E_y(\mathbf{x})$ and the domain specific content $\mathbf{c} = E_y(\mathbf{x})$ from an given image $\mathbf{x}$, which reflect the characteristics of reference images instead of randomly sampled noise.

**Foreground/Background (FG/BG) Disentanglement.** The generator $G$ (Figure 2(a)) translates an input image $\mathbf{x}$ into an output image $G(\mathbf{x}, \widetilde{\mathbf{s}}, \widetilde{\mathbf{c}})$ according to given domain specific style code $\widetilde{\mathbf{s}}$ and content $\widetilde{\mathbf{c}}$, which are provided either by the mapping network $M$ when generating from random noise or by the style-content encoder $E$ when transferring an existing content from a reference image. To achieve a **FG/BG** disentanglement, we split the channels of the three-dimensional feature map (i.e., $H \times W \times C$) at the bottle neck of $G$ into two parts. The model is then forced to encode the background into the first channels and the domain specific content $\hat{\mathbf{c}}$ into the latter channels by classification losses as discussed in Section 3.2. $\hat{\mathbf{c}}$ can then be replaced with content $\widetilde{\mathbf{c}}$ from the target domain. The adaptive instance normalization (AdaIN) (Huang & Belongie, 2017) is then used to inject $\widetilde{\mathbf{s}}$ into $\widetilde{\mathbf{c}}$ during the decoding process while the background $BG_G(\mathbf{x})$ is decoded separately. StarGAN v2 learns **FG** and **BG** together which leads to a conditional relationship between both. Our disentanglement and separate encoding break this conditioning and therefore enable our method to freely combine **FG** and **BG** as well as learn the full variation of **FG** content. Finally, $BG_G(\mathbf{x})$ and $\widetilde{\mathbf{c}}$ are concatenated together and then fused before output.

**Multi-task discriminator with auxiliary classifiers.** The discriminator $D$ (Figure 2(d)) is a multi-task discriminator with two auxiliary classifiers: a foreground domain classifier and a background domain classifier. This feature strengthens the disentanglement of **FG** and **BG** by first ensuring the input image $\mathbf{x}$ contains a domain specific content that can be recognized by the foreground domain classifier independent of the background. Later, each branch $D_y$ in the multi-task discriminator $D$ is trained to determine if an image $\mathbf{x}$ is a real image of its foreground domain or a fake image $G(\mathbf{x}, \mathbf{s}, \mathbf{c})$ generated by $G$. Apart from that, one extra branch $BG_{\text{cls}}$ is attached to decide whether the background information of the input images is well preserved.

**Content Transfer.** Mokady et al. (2020) introduced a concept that a model should be able to identity the difference between **two** domains when one of the domains contains a feature that the other does not have. We refer to this concept as 'anchor' and extend to multiple domains ($>2$) by the **FG/BG** disentanglement, the multi-task discriminator and the foreground content classifier in $D$. We treat domain Normal as the anchor domain i.e. set the domain specific content to zero, because a normal image has no domain specific content in our definition. As a result, we can now transfer contents between all combination of **FG** and **BG** domains (see Figure 10).

Compared to StarGAN v2, our method not only models style codes and contents separately but also disentangles the foreground and background of an image in a weakly-supervised manner. These features allow explicit control over output images by combining desired style codes and contents from one of the subnetworks with the input images. Therefore, it leads to higher variance regarding the location, structural pattern and artistic style of defects in the synthetic images of DT-GAN.

## 3.2 TRAINING OBJECTIVES

Given an image $\mathbf{x} \in \mathcal{X}$, its original foreground domain $y \in \mathcal{Y}$ and its background domain $p \in \mathcal{P}$, the following objectives are used to train our framework.

**Adversarial loss.** In the training phase, a noise vector $\mathbf{z} \in \mathcal{Z}$ and a target foreground domain $\widetilde{y} \in \mathcal{Y}$ are sampled randomly. Both of them are fed to $M$, producing a target style code $\widetilde{\mathbf{s}}$ and a target content $\widetilde{\mathbf{c}}$ as follows: $\widetilde{\mathbf{s}}, \widetilde{\mathbf{c}} = M_{\widetilde{y}}(\mathbf{z})$. Goal of the training is to ensure that $\widetilde{\mathbf{s}}$ and $\widetilde{\mathbf{c}}$ are sampled from the distribution over styles and contents of the target domain $\widetilde{y}$. The generator $G$ then combines an image $\mathbf{x}$ with $\widetilde{\mathbf{s}}$ and $\widetilde{\mathbf{c}}$ and learns to generate an output image $G(\mathbf{x}, \widetilde{\mathbf{s}}, \widetilde{\mathbf{c}})$ that is indistinguishable from real images in the target domain $\widetilde{y}$. We encourage this behavior by using an adversarial loss same as in Choi et al. (2020)

$$\mathcal{L}_{\text{adv}} = \mathbb{E}_{\mathbf{x},y}\big[\log D_y(\mathbf{x})\big] + \mathbb{E}_{\mathbf{x},\widetilde{y},\mathbf{z}}[\log\left(1 - D_{\widetilde{y}}(G(\mathbf{x}, \widetilde{\mathbf{s}}, \widetilde{\mathbf{c}}))\right)] \,, \tag{1}$$

where $D_y$ and $D_{\widetilde{y}}$ are the output branches of $D$ that correspond to the source domain $y$ and the target domain $\widetilde{y}$, respectively.

**Style-content reconstruction loss.** Similar to StarGAN v2, to enforce the generator $G$ takes the style code $\widetilde{\mathbf{s}}$ and the domain specific content $\widetilde{\mathbf{c}}$ into consideration during the generation process, we employ a style-content reconstruction loss

$$\mathcal{L}_{\text{sty\_con}} = \mathbb{E}_{\mathbf{x},\widetilde{y},\mathbf{z}}\big[\|\widetilde{\mathbf{s}} - S_E(G(\mathbf{x}, \widetilde{\mathbf{s}}, \widetilde{\mathbf{c}}))\|_1\big] + \mathbb{E}_{\mathbf{x},\widetilde{y},\mathbf{z}}\big[\|\widetilde{\mathbf{c}} - C_E(G(\mathbf{x}, \widetilde{\mathbf{s}}, \widetilde{\mathbf{c}}))\|_1\big] \,. \tag{2}$$

This objective urges the style-content encoder $E$ to recover $\widetilde{\mathbf{s}}$ and $\widetilde{\mathbf{c}}$ from $G(\mathbf{x}, \widetilde{\mathbf{s}}, \widetilde{\mathbf{c}})$. Here, the style-content encoder $E$ learns a mapping from an image to its style and content domains, which allows $G$ to synthesize an image with given $\mathbf{s}$ and $\mathbf{c}$ from reference images at test time.

**Diversity loss.** In order to further boost the diversity of output images from $G$, we introduce a loss that encourages diversity as follows: for a pair of random latent codes $\mathbf{z}_1$ and $\mathbf{z}_2$ we compute $\widetilde{\mathbf{s}}_i, \widetilde{\mathbf{c}}_i = M_{\widetilde{y}}(\mathbf{z}_i)$ for $i \in \{1, 2\}$ and enforce a different outcome of the generator $G$ for differently mixed style and content input pairs:

$$\begin{aligned} \mathcal{L}_{\text{ds}} = \ &\mathbb{E}_{\mathbf{x},\widetilde{y},\mathbf{z}_1,\mathbf{z}_2}\big[\|G(\mathbf{x}, \widetilde{\mathbf{s}}_1, \widetilde{\mathbf{c}}_2) - G(\mathbf{x}, \widetilde{\mathbf{s}}_2, \widetilde{\mathbf{c}}_1)\|_1\big] \\ &+ \mathbb{E}_{\mathbf{x},\widetilde{y},\mathbf{z}_1,\mathbf{z}_2}\big[\|G(\mathbf{x}, \widetilde{\mathbf{s}}_1, \widetilde{\mathbf{c}}_1) - G(\mathbf{x}, \widetilde{\mathbf{s}}_2, \widetilde{\mathbf{c}}_2)\|_1\big] \\ &+ \Sigma_{m,n,o}\big[\mathbb{E}_{\mathbf{x},\widetilde{y},\mathbf{z}_1,\mathbf{z}_2}\big[\|G(\mathbf{x}, \widetilde{\mathbf{s}}_m, \widetilde{\mathbf{c}}_n) - G(\mathbf{x}, \widetilde{\mathbf{s}}_o, \widetilde{\mathbf{c}}_o)\|_1\big]\big] \,, \end{aligned} \tag{3}$$

where $m, n \in \{1, 2 | m \neq n\}$ and $o \in \{1, 2\}$. Driven by this term, the generator $G$ is forced to discover meaningful style features and contents that eventually lead to diversity in generated images.

We ignore the denominator $\|\mathbf{z}_1 - \mathbf{z}_2\|_1$ of the original diversity loss (Mao et al., 2019a) for stable training as in StarGAN v2.

**Cycle consistency loss.** To ensure that the generated image $G(\mathbf{x}, \widetilde{\mathbf{s}}, \widetilde{\mathbf{c}})$ preserves the domain-invariant properties of its input image $\mathbf{x}$, we impose the cycle consistency loss (Zhu et al., 2017a)

$$\mathcal{L}_{\text{cyc}} = \mathbb{E}_{\mathbf{x},y,\widetilde{y},\mathbf{z}}\big[\|\mathbf{x} - G(G(\mathbf{x}, \widetilde{\mathbf{s}}, \widetilde{\mathbf{c}}), \hat{\mathbf{s}}, \hat{\mathbf{c}})\|_1\big] , \tag{4}$$

where $\hat{\mathbf{s}}, \hat{\mathbf{c}} = E_y(\mathbf{x})$ is the extracted style code and domain specific content of the input image $\mathbf{x}$, and $y$ is the original domain of $\mathbf{x}$. By learning to reconstruct the input image $\mathbf{x}$ with given style code $\hat{\mathbf{s}}$ and content $\hat{\mathbf{c}}$, the generator $G$ is then further encouraged to disentangle the background, the domain specific content and the style code.

**Content consistency loss.** Besides the cycle consistency loss, we apply another constraint to enforce that the detached domain specific content from $G$ is consistent with the one retrieved from $E$ according to

$$\mathcal{L}_{\text{con\_cyc}} = \mathbb{E}_{\mathbf{x},y,\widetilde{y},\mathbf{z}}\big[\|FG_G(\mathbf{x}) - \hat{\mathbf{c}}\|_1\big] + \mathbb{E}_{\mathbf{x},y,\widetilde{y},\mathbf{z}}\big[\|FG_G(G(\mathbf{x}, \widetilde{\mathbf{s}}, \widetilde{\mathbf{c}})) - \widetilde{\mathbf{c}}\|_1\big] , \tag{5}$$

where $\hat{\mathbf{c}} = E_y(\mathbf{x})$, $\widetilde{\mathbf{c}} = E_{\widetilde{y}}(\mathbf{x})$, $FG_G(\mathbf{x})$ and $FG_G(G(\mathbf{x}, \widetilde{\mathbf{s}}, \widetilde{\mathbf{c}}))$ are the pop-out domain specific content from input image $\mathbf{x}$ and generated image $G(\mathbf{x}, \widetilde{\mathbf{s}}, \widetilde{\mathbf{c}})$, respectively.

**Classification losses.** We employ two classification losses: the first one is the foreground content classification loss

$$\mathcal{L}_{\text{FG\_cls}} = \mathbb{E}_{\mathbf{x}_{\text{real}},y}\big[-\log D_{\text{FG\_cls}}(y|\mathbf{x}_{\text{real}})\big] + \mathbb{E}_{\mathbf{x}_{\text{fake}},\widetilde{y}}\big[-\log D_{\text{FG\_cls}}(\widetilde{y}|\mathbf{x}_{\text{fake}})\big] , \tag{6}$$

which aims to ensure that the domain specific content is properly encoded and carries enough information from the target domain. The second one is the background classification loss

$$\mathcal{L}_{\text{BG\_cls}} = \mathbb{E}_{\mathbf{x}_{\text{real}},p}\big[-\log D_{\text{BG\_cls}}(p|\mathbf{x}_{\text{real}})\big] + \mathbb{E}_{\mathbf{x}_{\text{fake}},p}\big[-\log D_{\text{BG\_cls}}(p|\mathbf{x}_{\text{fake}})\big] , \tag{7}$$

where $p$ is the corresponding background type of $\mathbf{x}_{\text{real}}$ and $\mathbf{x}_{\text{fake}}$. With the help of this objective, the generator $G$ learns to preserve the domain-invariant characteristics of its input image $\mathbf{x}$ while dissociating the foreground domain specific part.

**Full objective.** Our full objective functions can be summarized as

$$\min_{G,F,E} \max_{D} \quad \mathcal{L}_{\text{adv}} + \lambda_{\text{sty\_con}} \mathcal{L}_{\text{sty\_con}} - \lambda_{\text{ds}} \mathcal{L}_{\text{ds}} + \lambda_{\text{cyc}} \mathcal{L}_{\text{cyc}} +$$
$$\lambda_{\text{con\_cyc}} \mathcal{L}_{\text{con\_cyc}} + \lambda_{\text{FG\_cls}} \mathcal{L}_{\text{FG\_cls}} + \lambda_{\text{BG\_cls}} \mathcal{L}_{\text{BG\_cls}} , \tag{8}$$

where $\lambda_{\text{sty}}, \lambda_{\text{ds}}, \lambda_{\text{cyc}}, \lambda_{\text{con\_cyc}}, \lambda_{\text{FG\_cls}}$ and $\lambda_{\text{BG\_cls}}$ are the hyperparameters for each term.

## 4 Experiments

We evaluated the images generated by DT-GAN through a series of experiments both quantitatively and qualitatively. Finally, we demonstrate the benefits of our generated images when being used as data augmentation for a defect classification task on limited data.

**Dataset.** All experiments were performed on a real industrial dataset: a Surface Defect Inspection (SDI) dataset that contains three different kinds of products from production lines and samples from each product are classified into three mutually exclusive classes: Normal, Scratch and Spot. All of the images are grayscale. Detailed statistics of the dataset are summarized in Appendix A. Note that only the training set was used in GAN training, the test set was left untouched for final evaluation in classifier training. For a fair comparison, all images were resized to $128 \times 128$ resolution for both GAN training and classifier training, which was also the highest resolution used in the baselines for image generation. For comparison, we also conducted experiments on the widely used MVTec Anomaly Detection dataset (Bergmann et al., 2019) in Appendix E.4.

### 4.1 Defect Generation

**Baselines.** As discussed in Section 3, DT-GAN can either use the mapping network to randomly generate styles and defects, or it can use the style-content encoder to extract both from reference images. We refer to these cases as 'latent-guided' and 'reference-guided', respectively.

Since the two ways of guidance are fundamentally different, we evaluated them against two sets of baselines: Our reference-guided image generation was compared to Mokady et al. (2020) and StarGAN v2, because both of them can perform a reference-guided translation. Note that Mokady et al. (2020) can only translate between two domains while StarGAN v2 and DT-GAN can achieve multi-domain translation within a single model. Images generated through the latent-guided part of DT-GAN were compared to state-of-the-art GANs in image synthesis: BigGAN (Brock et al., 2019) and StyleGAN v2 (Karras et al., 2020b). We set BigGAN to condition on defect types during training while StyleGAN v2 was trained unconditionally. All baselines were trained from scratch with the public implementations provided by the authors[1].

### 4.1.1 QUANTITATIVE EVALUATION

**Metrics.** We employed the commonly used frechet inception distance (FID) (Heusel et al., 2017) to evaluate both the visual quality and the diversity of the generated images. We also report the kernel inception distance (KID) (Binkowski et al., 2018) which is a more stable metric for small sets of images like our SDI dataset. Lower FID and KID scores indicate better performance.

Both scores are shown in Table 1. We observe that methods like BigGAN and StyleGAN v2, which perform defect synthesis purely based on latent codes, generally provide unsatisfactory results on the SDI dataset, presumably due to the small number of defective samples that were available. These methods then struggle to capture the complex and irregular patterns of defects. We also experimented with augmentation methods for GAN training (Karras et al., 2020a; Zhao et al., 2020) but did not find a consistent improvement (see Appendix E.2). We thus only report the best scores.

Reference-guided synthesis methods like Mokady et al. (2020) and StarGAN v2 seem to generate more realistic images. The scores of StarGAN v2 on a single product are omitted here because generating images with specified background is not possible due to its network design—the product type changes in output images, which we refer to as 'identity-shift'. As seen in Table 1, our method achieves better scores in all cases. We believe this is due to the fact that our method allows free combination of foreground defects and backgrounds, making the generated images more diverse even with a small number of training samples.

Table 1: Quantitative comparison of DT-GAN with baseline image synthesis methods using FID and KID. Note that the reported values are not comparable between columns, because they were calculated on different training sets.

| Method | FID↓ | | | | KID↓ | | | |
|---|---|---|---|---|---|---|---|---|
| | A | B | C | All | A | B | C | All |
| Mokady (2020) | 68.69 | 66.90 | 36.21 | 58.63 | 0.050 | 0.036 | 0.030 | 0.036 |
| StarGAN v2 | - | - | - | 37.70 | - | - | - | 0.013 |
| StyleGAN v2 | 90.10 | 52.95 | 138.09 | 35.34 | 0.072 | 0.027 | 0.186 | 0.013 |
| BigGAN + DiffAug | 218.74 | 134.41 | 270.89 | 155.88 | 0.220 | 0.121 | 0.378 | 0.099 |
| Ours | **58.43** | **36.44** | **22.68** | **29.73** | **0.025** | **0.013** | **0.012** | **0.009** |

### 4.1.2 QUALITATIVE EVALUATION

We present a qualitative comparison with the baseline methods in latent-guided image synthesis in Figure 3. To make a fair comparison, we trained StyleGAN v2 and BigGAN on each product separately to have control on background products. Note however, that images from DT-GAN were always obtained from a single model. We can see that some generated samples from StyleGAN v2 do not contain clear defects, and samples from BigGAN present abnormal grid patterns. Both methods do not take images as inputs but generate synthetic images according to a given latent code which contains information for both **FG** and **BG**. This conditioning leads to limited diversity in the output images. On the other hand, StarGAN v2 performs translation based on input images but suffers from the same entanglement issue. Thus, it fails to preserve the background, which results in artifacts or identity-shift in its outputs. Our network architecture that disentangles foreground and background seems to mitigate these issues. See Appendix E.4 for more images.

---

[1]We could not obtain the code of Defect-GAN to reproduce their results.

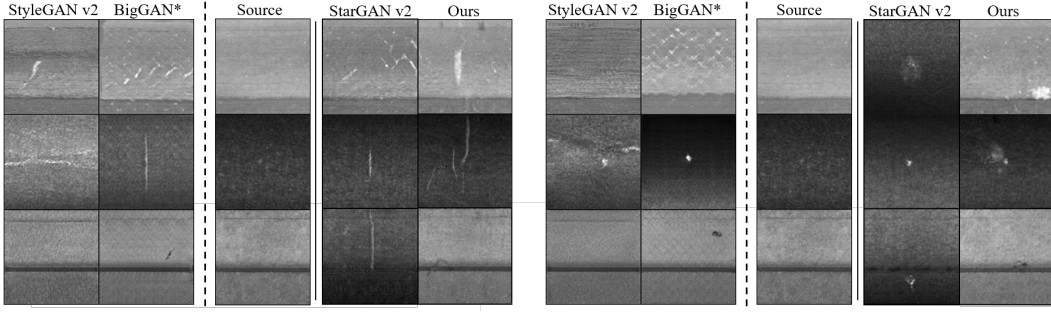

(a) Normal-to-Scratches  (b) Normal-to-Spots

Figure 3: Qualitative comparison of latent-guided image synthesis results. In each subfigure: on the left, defective images are fully generated from random noise. On the right, random defects are synthesized onto given normal samples. Note that BigGAN* denotes it was trained with DiffAug.

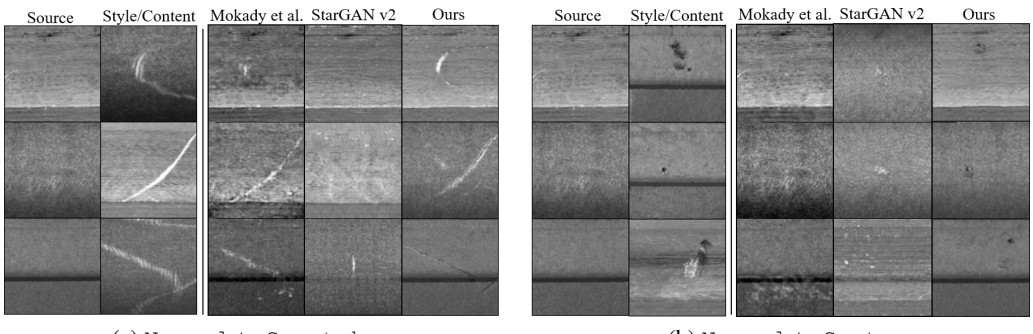

(a) Normal-to-Scratches  (b) Normal-to-Spots

Figure 4: Qualitative comparison of reference-guided image synthesis results on the SDI dataset. Each method transforms the given source images into target foreground domains (e.g., Scratches) with the styles and contents extracted from the reference images.

Also for reference-guided image synthesis, where we used different background and foreground reference images as illustrated in Figure 4, only our method produces high quality images with preserved background from the source and transferred foreground defect from the reference.

**Ablation study.** We visually demonstrate the effect of each component we added to DT-GAN compared to StarGAN v2 in Figure 5, using the examples of both latent- and reference-guided image synthesis from Normal to Scratches. The quantitative evaluation can be found in Appendix E.3.

Column (a) corresponds to StarGAN v2 and highlights the drawback of entangled **FG/BG** again (i.e. the identity-shift in the background). We first tackle this problem by modeling the style code and foreground content explicitly and feeding them separately to the generator. This leads to a better preservation of the background structure in column (b) for the reference-guided subnetwork, but not for the latent-guided synthesis on the bottom of Figure 5. Thus, we add a foreground classifier in the discriminator in (c) to ensure the output image contains the desired foreground content (scratch). Similarly, we introduce a background classifier to the discriminator in column (d). Note that the additional product type labels can be acquired automatically from production lines.

For column (e), we add the separate decoders for foreground and background in the generator which are fused only in the end. This enhances the preservation of background characteristics like lighting even more. Imposing an additional penalty for foreground content extracted from a normal sample as described in Section 3.1 leads to another visual improvement of the foreground edges for reference-guided synthesis in column (f). Finally, inspired by StyleGAN, we incorporate adaptive noise injection to the mapping network, which significantly boosts the performance of our latent-guided image synthesis as shown in column (g).

**Styling.** We visually demonstrate the effect of style codes in our method by randomly sampling those and combining them with fixed reference background and foreground images in Figure 6, where a variety of artistic styles can be seen on the output columns.

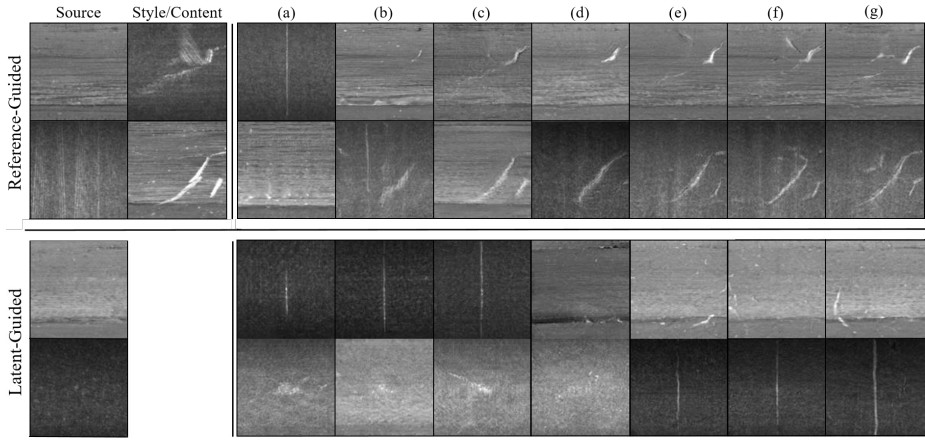

Figure 5: Ablation study. (a) The baseline StarGAN v2. (b) + Style-Content branches. (c) + Foreground classifier. (d) + Background classifier. (e) + Separately decoding foreground and background in $G$. (f) + Anchor foreground domain (e.g. `Normal`). (g) + Noise injection in Mapping Network.

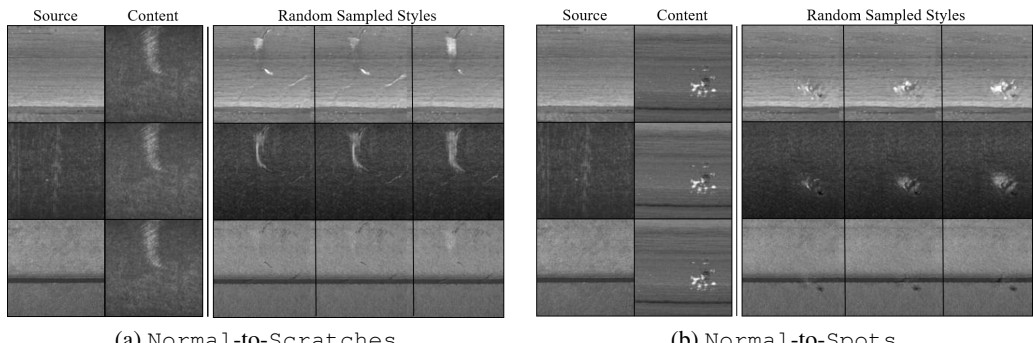

(a) `Normal`-to-`Scratches`                    (b) `Normal`-to-`Spots`

Figure 6: Visual effect of randomly sampled style codes on fixed pairs of reference background (Source) and foreground (Content) images.

## 4.2 DT-GAN FOR DATA AUGMENTATION

We also evaluated our method as a data augmentation method for defect classification on the SDI dataset. We defined one task 'general', where the classifier was trained on images from all products at once, while task 'single product' only used the subset of images for one product.

Besides, we incrementally varied the amount of real `Normal` data available for classifier training: 4500, 6600, 12000 and 18600. In the case of defective images, all of them were always used due to the small amount unless otherwise specified. As backbone we used a ResNet-50 (He et al., 2016a) with ImageNet pretrained weights. For experiments with synthetic data, we attached an auxiliary domain classifier to the network through a Gradient Reversal Layer (Ganin & Lempitsky, 2015).

Table 2: Quantitative comparison of the baseline methods on defect classification task at the scale of 12000 images/class. The reported values are the achieved error rates (%) over five runs.

| Method | ResNet-50 | EfficientNet-b4 |
|---|---|---|
| No-Aug | 21.64±1.24 | 12.06±0.64 |
| Trad-Aug | 12.58±0.81 | 9.33±0.73 |
| Mokady (2020) | 11.11±1.19 | 13.26±1.13 |
| StarGAN v2 | 13.07±1.30 | 12.25±0.79 |
| StyleGAN v2 | 11.55±1.79 | 11.68±0.76 |
| BigGAN+DiffAug | 11.45±0.61 | 12.06±0.50 |
| Ours | **9.9±0.69** | **9.14±1.02** |

Since the SDI dataset is highly imbalanced, we oversampled the minority classes (Ling et al., 1998) unless the data was balanced through synthetic images. Additionally, we always applied traditional data augmentation techniques like random horizontal flips, jittering and lighting (Shorten & Khosh-goftaar, 2019) except where noted. All following results were evaluated by the achieved error rates over five runs with different random seeds.

**Effectiveness of synthetic data.** We first compare classifier performance for no augmentation (**No-Aug**), traditional data augmentation (**Trad-Aug**), and a combination of traditional augmentation with synthetic images for GAN methods including DT-GAN. We also introduce a stronger back-bone, EfficientNet-b4 (Tan & Le, 2019), to demonstrate that our results are not confined to a specific network. Table 2 shows that our method is the only one that improves performance for both back-bones, presumably due to the combination of high visual image quality and diversity in our samples.

Table 3: Experimental results on using different amount of synthetic images generated by DT-GAN to train classifiers. The left-most column stands for number of samples per class to be classified. The training set of the baselines is balanced by oversampling while ours is by synthetic images.

| Dataset Size | 20A | | All | |
| --- | --- | --- | --- | --- |
| | Trad-Aug | Ours | Trad-Aug | Ours |
| 4500 | 15.55±0.63 | **14.28±1.25** | 12.75±0.61 | **11.04±0.76** |
| 6600 | 16.69±0.76 | **14.41±3.12** | 13.07±1.57 | **10.60±0.48** |
| 12000 | 16.95±1.02 | **14.22±1.53** | 12.05±0.81 | **9.90±0.69** |
| 18600 | 16.12±2.19 | **15.36±0.86** | 12.37±0.32 | **10.21±0.96** |

**Impact of dataset size.** Motivated by the limited availability of data in real-world production sce-narios, we therefore evaluated DT-GAN for data augmentation on a subset of the full SDI dataset (**All**), which only contains 20 defective samples in product A for each defect type (**20A**). In this case, DT-GAN was also trained on the reduced subset. As shown in Table 3, there is a clear improvement when synthetic images from DT-GAN are used as data augmentation, even for the extremely limited data subset. Further results on single product classifiers can be found in Appendix E.1.

Table 4: Cross-domain effect on single product classifiers trained with reference-guided synthetic images at the scale of 12000 images/class.

| | Trad-Aug | vA | vB | vC | vABC |
| --- | --- | --- | --- | --- | --- |
| **A** | 13.81±2.36 | 11.81±2.65 | 12.72±2.87 | 11.99±1.63 | **11.09±3.49** |
| **B** | 6.80±1.64 | 6.40±1.34 | 6.60±1.52 | 6.59±1.34 | **5.60±1.34** |
| **C** | 16.57±3.20 | 13.14±2.81 | **11.23±0.80** | 14.85±1.73 | 11.42±0.96 |

**Cross-domain effect.** We hypothesized that limited data can be counteracted by transferring de-fects across multiple background products, if there are at least some defects that occur on multiple products (See Appendix E.1 for further discussion). We tested this approach by comparing the per-formance of classifiers trained on synthetic images with defects from a specific source (**vA**, **vB**, **vC**) to classifiers trained on images with defects from all products (**vABC**). As we can see in Table 4, the best performances are reached by the models that take over defects from other products. We interpret this as support for our hypothesis and its practical usefulness.

## 5 CONCLUSION

We propose a novel method, DT-GAN, which allows diverse defect synthesis both by generating from randomly sampled noise and by following the guidance of given reference images. Due to explicit style-content separation and **FG/BG** disentanglement, DT-GAN achieves higher image fidelity, better variance in defects and full control over background and foreground while being sample-efficient. We demonstrated the feasibility and benefits of DT-GAN on a real industrial defect classification task and the results show our method provides consistent gains even with limited data and boosts the performance of classifiers compared to state-of-the-art image synthesis methods. For future investigation, we aim to represent defects more explicitly (e.g., localization) to improve the explainability of the model and also enhance the model transferability to unseen products.

REPRODUCIBILITY STATEMENT

We aim for full reproducibility by publishing the source code and dataset with the final version of the paper. Besides, we provide descriptions of the training details in Appendix B, the evaluation setup in Appendix C and the network architecture in Appendix D.

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

## A  THE SURFACE DEFECT INSPECTION DATASET

The Surface Defect Inspection (SDI) dataset consists of 20,414 images at $128 \times 128$ resolution. It contains three background domains—product **A**, product **B** and product **C**, each can be further classified into three foreground domains—`Normal`, `Scratches` and `Spots`. Figure 7 shows example images of the SDI dataset. To be noticed that the dataset is highly imbalanced not only between normal and defective samples but also between different products as shown in Table 5. This sets a more challenging task when training deep neural networks like GANs and downstream classifiers.

For each foreground and background domains, we randomly select 50 images for a joint validation/test set, which is then further split into separate sets in the ratio of 3:7, and use all remaining images as training sets for GAN and classifier training. We present the distribution of the training set when training DT-GAN in Table 6. Note that the normal samples used in GAN training are only a subset of all available samples in `Normal` and we keep the rest of them for generating defective samples at test time. For classifier training, we show the statistics in Table 7, where the number of normal samples involved in classifier training increase incrementally. The validation set is used to select the best model during classifier training while the test set is left untouched until the final evaluation. Both of the validation and test set are inaccessible by DT-GAN.

Table 5: Distribution of the full SDI dataset.

|  | **Overview** | | |
| --- | --- | --- | --- |
|  | **A** | **B** | **C** |
| Normal | 6250 | 6250 | 6250 |
| Scratches | 340 | 167 | 121 |
| Spots | 108 | 670 | 258 |

Table 6: The training set for DT-GAN and the baseline image synthesis methods.

|  | **Overview** | | |
| --- | --- | --- | --- |
|  | **A** | **B** | **C** |
| Normal | 700 | 700 | 700 |
| Scratches | 290 | 117 | 71 |
| Spots | 58 | 620 | 208 |

Table 7: The training, validation and test set for classifier training, where $N$ increases incrementally—1500, 2200, 4000 and 6200.

|  | **Train** | | | **Validation** | | | **Test** | | |
| --- | --- | --- | --- | --- | --- | --- | --- | --- | --- |
|  | **A** | **B** | **C** | **A** | **B** | **C** | **A** | **B** | **C** |
| Normal | $N$ | $N$ | $N$ | 12 | 18 | 15 | 38 | 32 | 35 |
| Scratches | 290 | 117 | 71 | 14 | 16 | 15 | 36 | 34 | 35 |
| Spots | 58 | 620 | 208 | 14 | 16 | 15 | 36 | 34 | 35 |

## B  TRAINING DETAILS

**DT-GAN.** We follow the training scheme as described in StarGAN v2 with minor modifications. To fit the model on a single Nvidia GTX TITAN X, the batch size is reduced to four while the model is still trained for 100,000 iterations. The training time is about three and a half days on the dedicated GPU with the modified network architecture[2] and loss functions mentioned in Section 3 in PyTorch (Paszke et al., 2017). We set $\lambda_{sty} = 1$, $\lambda_{ds} = 1$, $\lambda_{cyc} = 1$, $\lambda_{con\_cyc} = 1$, $\lambda_{cls} = 1$ and $\lambda_{BG\_cls} = 1$ for the SDI dataset. All other design choices remain the same as in StarGAN v2.

**Classifiers.** We train all the classifiers that use ResNet-50 as backbone for 100 epochs with the SGD optimizer (Ruder, 2016) and batch size 256. The initial learning rate is 0.001, momentum is 0.9 and weight decay is 1e-4. A learning rate scheduler is set to reduce the learning rate by factor of 0.1 when the validation loss stops decreasing for 5 epochs. The same setting also applies to EfficientNet-b4, except the batch size is reduced to 128. Although DT-GAN can synthesize realistic defective samples, we notice that there still exists a domain gap between the generated samples and the real samples. To explore the full potential of the generated samples, we attach an auxiliary

---

[2]We based our implementation on source code from StarGAN v2: https://github.com/clovaai/stargan-v2

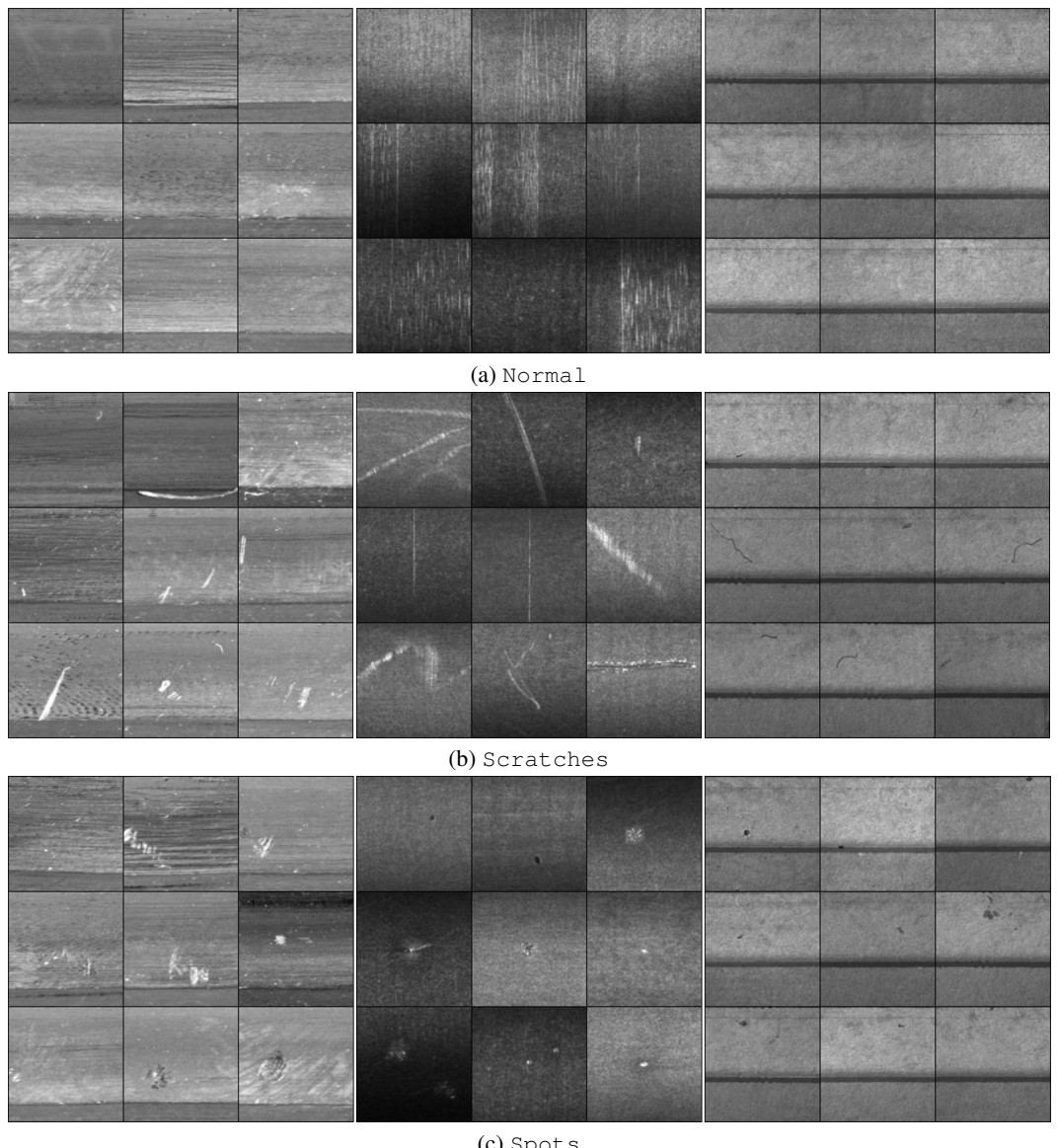

(a) `Normal`

(b) `Scratches`

(c) `Spots`

Figure 7: Overview of the SDI dataset.

source classifier to distinguish between synthetic and real samples. Then, this classifier is connected to the backbone (e.g. ResNet-50) through a Gradient Reversal Layer. With the help of the Gradient Reversal Layer, the backbone is forced to extract the shared features between synthetic and real samples, which ensures all training samples are effectively learned.

We design a two-layer perceptron that connects to the average pooling layer in ResNet-50 as shown in Figure 8. Note that the usual fully connected layer after the average pooling in ResNet-50 remains the same and is not affected by the extra branch we added. Inspired by Chen et al. (2018), a three-layer perceptron is used for EfficientNet-b4 instead as shown in Figure 9. Its layers are initialized with a random normal distribution, where the standard deviation is set to 0.01 for the first two layers and 0.05 for the output layer. The biases for all layers are set to 0.

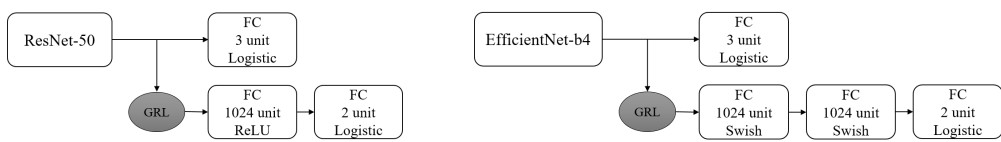

Figure 8: ResNet-50 with GRL.                    Figure 9: EfficientNet-b4 with GRL.

## C  EVALUATION SETUP

**Generated samples from DT-GAN.** DT-GAN requires images as input for generating synthetic data. At test time, we translated each `Normal` image in the SDI dataset into four defective images: two with `Scratches` and two with `Spots`. The translations were performed by two subnetworks: by the mapping network $M$ using random noise ('latent-guided') and by the style-content encoder $E$ using a reference image ('reference-guided'). We first randomly sampled one latent code for each defective foreground domain. Similarly, we also randomly sampled one reference image from the training set for each defective foreground domain. The corresponding style codes and defect contents were then produced by the two subnetworks respectively and fed to the generator for target image generation.

We conducted classification experiments separately on images generated from the two subnetworks and a mixture set of both (i.e. 50% from each subnetwork). Experiments show consistent gains of using synthetic images generated from DT-GAN (Table 8). We observe that the latent-guided synthetic images in general perform better than the reference-guided one, while the mixture set provides more stable results with regard to the standard deviation. Presumably the mixture set benefits from the combination of samples from reference-guided synthesis, which are well aligned with the original defect distribution, and the samples from latent-guided synthesis, i.e. from random noise, which adds novel but plausible defects to the dataset. In the main text, we report the results of the mixture set for all experiments, including the quantitative evaluation of DT-GAN.

Table 8: Classification results with regard to the synthetic images generated from the two subnetworks and the mixture set.

| Dataset Size | All | | | |
|---|---|---|---|---|
| | Trad-Aug | Latent | Reference | Mix |
| 4500 | 12.75±0.61 | **10.72±0.96** | 11.48±0.88 | 11.04±0.76 |
| 6600 | 13.07±1.57 | **10.34±1.86** | 11.55±1.64 | 10.60±0.48 |
| 12000 | 12.05±0.81 | 9.90±1.26 | 10.40±0.99 | **9.90±0.69** |
| 18600 | 12.37±0.32 | 11.04±1.26 | 12.12±0.75 | **10.21±0.96** |

**Frechét inception distance (FID) and Kernel inception distance (KID).** We used the feature vectors from the last average pooling layer of the ImageNet pretrained Inception-V3 to calculate both scores. For each test image from the `Normal` domain, we translated it into a synthetic defective image of each defect domain. The style codes and contents for the translation were acquired in two ways: by randomly sampling from the standard normal distribution and by randomly sampling a reference image from the train set of a defect domain. To calculate the FID and KID score, we generated 4000 defective samples per product per defect domain for each way of guidance, and formed the mixture set by randomly sampling 2000 images per product per defect domain from each way. The reported FID and KID scores were then computed between the defective images in the training set and the mixture set of synthetic defective images. The same procedure was applied when computing scores on single product subsets of the SDI dataset. For example, for product A, we calculated the scores between the defective image of product A in the training set and the mixture set of synthetic defective images of product A.

## D  NETWORK ARCHITECTURE

In this section, we provide the architectural details of all four modules in DT-GAN.

Table 9: Generator architecture.

(a) Encoder

| Layer | Resample | Norm | Output Shape |
|---|---|---|---|
| Image $\mathbf{x}$ | - | - | $128 \times 128 \times 3$ |
| Conv $1\times1$ | - | - | $128 \times 128 \times 128$ |
| ResBlk | AvgPool | IN | $64 \times 64 \times 256$ |
| ResBlk | AvgPool | IN | $32 \times 32 \times 512$ |
| ResBlk | AvgPool | IN | $16 \times 16 \times 512$ |
| ResBlk | - | IN | $16 \times 16 \times 512$ |
| ResBlk | - | IN | $16 \times 16 \times 512$ |

(b) Background Decoder

| Layer | Resample | Norm | Output Shape |
|---|---|---|---|
| Input | - | - | $16 \times 16 \times 448$ |
| ResBlk | - | IN | $16 \times 16 \times 448$ |
| ResBlk | - | IN | $16 \times 16 \times 512$ |
| ResBlk | - | IN | $16 \times 16 \times 512$ |
| ResBlk | Upsample | IN | $32 \times 32 \times 512$ |
| ResBlk | Upsample | IN | $64 \times 64 \times 256$ |
| ResBlk | Upsample | IN | $128 \times 128 \times 448$ |

(c) Foreground Decoder

| Layer | Resample | Norm | Output Shape |
|---|---|---|---|
| Input | - | - | $16 \times 16 \times 64$ |
| ResBlk | - | AdaIN | $16 \times 16 \times 64$ |
| ResBlk | - | AdaIN | $16 \times 16 \times 256$ |
| ResBlk | - | AdaIN | $16 \times 16 \times 256$ |
| ResBlk | Upsample | AdaIN | $32 \times 32 \times 256$ |
| ResBlk | Upsample | AdaIN | $64 \times 64 \times 128$ |
| ResBlk | Upsample | AdaIN | $128 \times 128 \times 64$ |

(d) Fusion

| Layer | Resample | Norm | Output Shape |
|---|---|---|---|
| Input | - | - | $128 \times 128 \times (448 + 64)$ |
| Conv $1\times1$ | - | - | $128 \times 128 \times 3$ |

Table 10: Mapping network architecture.

(a) Shared Layers

| Layer | Activation | Output Shape |
|---|---|---|
| Latent $\mathbf{z}$ | - | 16 |
| Linear | ReLU | 512 |
| Linear | ReLU | 512 |
| Linear | ReLU | 512 |
| Linear | ReLU | 512 |

(b) Style Code

| Layer | Activation | Output Shape |
|---|---|---|
| Input | - | 512 |
| Linear | ReLU | 512 |
| Linear | ReLU | 512 |
| Linear | ReLU | 512 |
| Linear | - | 64 |

(c) Content

| Layer | Resample | Activation | Noise | Output Shape |
|---|---|---|---|---|
| Input | - | - | - | 512 |
| Reshape | - | - | - | $1 \times 1 \times 512$ |
| ResBlk | Upsample | IN | True | $2 \times 2 \times 512$ |
| ResBlk | Upsample | IN | True | $4 \times 4 \times 512$ |
| ResBlk | Upsample | IN | True | $8 \times 8 \times 256$ |
| ResBlk | Upsample | IN | True | $16 \times 16 \times 128$ |
| Conv $1\times1$ | - | IN | True | $16 \times 16 \times 64$ |

**Generator** (Table 9). For the SDI dataset, the encoder part of the generator consists of three down-sampling blocks and two intermediate blocks (Table 9 (a)), all of them are pre-activation residual units (He et al., 2016b). Then the encoded feature map is split channel-wise into background (Table 9 (b)) and foreground (Table 9 (c)). Both of them are then carried through separate decoders. We use the instance normalization (IN) and the adaptive instance normalization (AdaIN) as indicated. The style code is injected into all AdaIN layers to modulate the affine transformations. Note that

Table 11: Style-content encoder and discriminator architectures.

(a) Shared Layers

| Layer | Resample | Norm | Output Shape |
|---|---|---|---|
| Input $\mathbf{x}$ | - | - | $128 \times 128 \times 3$ |
| Conv 1×1 | - | - | $128 \times 128 \times 64$ |
| ResBlk | AvgPool | - | $64 \times 64 \times 256$ |
| ResBlk | AvgPool | - | $32 \times 32 \times 512$ |
| ResBlk | AvgPool | - | $16 \times 16 \times 512$ |

(b) Style Code / Discriminator and BG Classifier

| Layer | Resample | Norm | Output Shape |
|---|---|---|---|
| Input | - | - | $16 \times 16 \times 512$ |
| ResBlk | AvgPool | - | $8 \times 8 \times 512$ |
| ResBlk | AvgPool | - | $4 \times 4 \times 512$ |
| LReLU | - | - | $4 \times 4 \times 512$ |
| Conv 4×4 | - | - | $1 \times 1 \times 512$ |
| LReLU | - | - | $1 \times 1 \times 512$ |
| Reshape | - | - | $512$ |
| Linear $*K$ | - | - | $D * K$ |

(c) Content / FG Classifier

| Layer | Resample | Norm | Output Shape |
|---|---|---|---|
| Input | - | - | $16 \times 16 \times 512$ |
| LReLU | - | - | $16 \times 16 \times 512$ |
| Conv 1×1 $*K$ | - | - | $16 \times 16 \times 64 *K$ |

AdaIN is only used in the foreground decoder. The outputs of both decoders are only fused in the end (Table 9 (d)).

**Mapping Network** (Table 10). The mapping network consists of four shared linear layers (Table 10 (a)) and two separate branches: one for generating style codes (Table 10(b)) and one for contents (Table 10(c)). Each of them is further divided into $K$ output branches, where $K$ denotes the number of domains. The dimension of the input, the output style code and the output content is set to 16, 64, and $16 \times 16 \times 64$, respectively. The latent code is sampled from the standard normal distribution. Note that we apply per-pixel noise after each convolution in the content branch, which we have observed to increase the diversity of generated defects significantly (cf. Figure 5 (g)).

**Style-Content Encoder** (Table 11). The style-content encoder consists of a CNN (Table 11 (a)) with two branches (Table 11 (b) and (c)) as in the mapping network. Each branch has $K$ outputs, where $K$ is the number of domains. Three pre-activation residual blocks are shared among two branches, followed by a specific structure for each branch. The output dimension $D$ in Table 11 is set to 64, which denotes the dimension of the style code.

**Discriminator** (Table 11). The discriminator is a multi-task discriminator with two auxiliary classifiers for the foreground content and the background. The structure is almost identical to the style-content encoder, except $D$ is set to 1 for real/fake classification. The background classifier acts in parallel to final linear layer in Table 11 (b) and provides the logits for background classification. The foreground classifier instead acts on top of the output in Table 11 (c) and four more pre-activation residual layers are applied to encode the content into logits for foreground content classification.

# E   ADDITIONAL RESULTS

## E.1   ADDITIONAL RESULTS ON THE SDI DATASET

We provide additional reference-guided image synthesis results on the SDI dataset in Figure 10. We demonstrate all the possible transfers among all foreground domains. Both style codes and contents are extracted from the reference images. To be noted that DT-GAN can append and remove foreground defects not only onto `Normal` samples but also to defective samples. For example, in the fifth column of Figure 10, the original scratch in the source image is removed and only the defects from the reference images are presented in the output images.

Besides, we present additional evaluations showing the effectiveness of our synthetic data according to Table 3. As seen in Table 12, the synthetic images from DT-GAN also boost the performance in

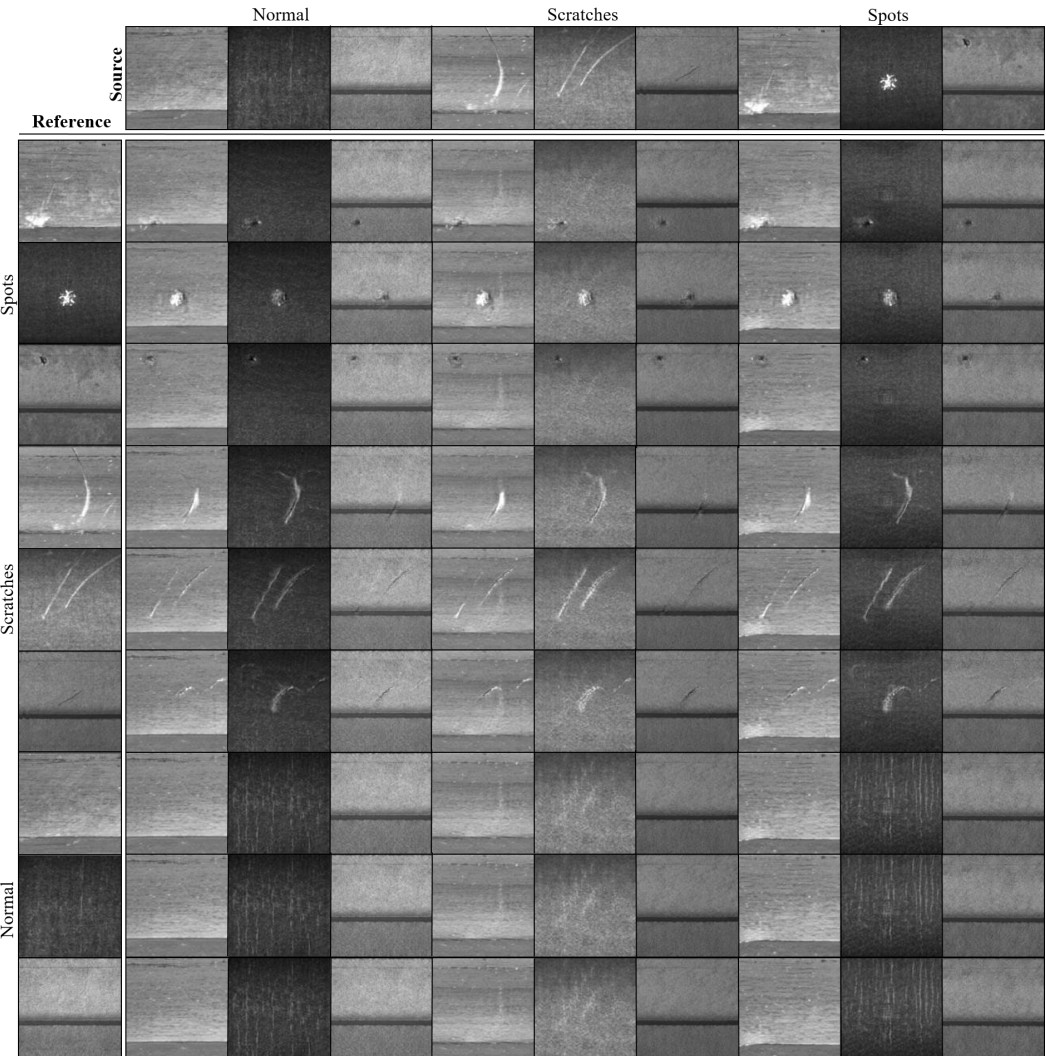

Figure 10: Reference-guided image synthesis results on the SDI dataset. The first row and the first column are the real images sampled from the dataset, while the rest are synthetic images generated by the proposed DT-GAN. Our model provides translations between different foreground domains (`Normal`, `Scratches` and `Spots`) with styles and contents extracted from reference images while the backgrounds from source images are well preserved.

single product classifiers, where the classifiers were trained on the subset of images for one product (**A**, **B**, **C**) instead of the full dataset (**ABC**).

As discussed in Section 4.2, we assumed that the data-insufficiency problem can be mitigated by transferring defects across multiple background products. To examine if this assumption holds, we compared the performance of classifiers trained on synthetic images with defects from a specific source (**vA**, **vB**, **vC**) to classifiers trained on images with defects from all products (**vABC**). The results on the cross-domain effect with regard to different sizes of the training set are shown in Table 13. We again notice that using our synthetic data is beneficial. Moreover, in most cases the performance is further improved by exploiting cross-domain information (i.e. by transferring defects from other products). We interpret this as support for our assumption and the practical usefulness of our method in the real-world scenario. The case of cross-domain image synthesis when the desired combination is not presented in the training set is covered in the study on the MVTec Anomaly Detection dataset (Bergmann et al., 2019) (see Appendix E.4).

Table 12: Quantitative results for DT-GAN as a data augmentation method to train general and single product classifiers. The left-most column indicates the number of samples per class, including all images from the training set plus increasing amounts of synthetic images. In the first row, **20A** refers to the case of 20 real defective samples for product A, while **All** refers to the full training set.

| Dataset Size | 20A | | | | | | | |
| --- | --- | --- | --- | --- | --- | --- | --- | --- |
| | A | | B | | C | | ABC | |
| | Trad-Aug | Ours | Trad-Aug | Ours | Trad-Aug | Ours | Trad-Aug | Ours |
| 4500 | 35.09±2.62 | **27.64±3.12** | 7.8±1.48 | **5.6±1.67** | 15.24±1.90 | **13.14±1.70** | 15.55±0.63 | **14.28±1.25** |
| 6600 | 39.64±2.28 | **27.64±1.65** | 8.8±1.64 | **6.2±1.64** | 15.81±1.73 | **12.38±1.65** | 16.69±0.76 | **14.41±3.12** |
| 12000 | 34.18±4.39 | **28.55±7.32** | 5.8±0.45 | **5.6±1.14** | 16.19±1.17 | **10.86±1.28** | 16.95±1.02 | **14.22±1.53** |
| 18600 | 39.45±7.06 | **32.55±5.04** | 7.2±0.84 | **5.2±1.10** | 14.86±0.85 | **13.14±2.06** | 16.12±2.19 | **15.36±0.86** |

| Dataset Size | All | | | | | | | |
| --- | --- | --- | --- | --- | --- | --- | --- | --- |
| | A | | B | | C | | ABC | |
| | Trad-Aug | Ours | Trad-Aug | Ours | Trad-Aug | Ours | Trad-Aug | Ours |
| 4500 | 16.00±1.04 | **10.18±1.75** | 8.79±0.45 | **5.60±1.51** | 17.13±6.62 | **14.09±2.27** | 12.75±0.61 | **11.04±0.76** |
| 6600 | 14.90±1.38 | **10.54±1.22** | 7.60±1.51 | **6.80±3.11** | 15.23±2.33 | **11.42±0** | 13.07±1.57 | **10.60±0.48** |
| 12000 | 13.81±2.36 | **6.72±1.65** | 6.80±1.64 | **4.60±0** | 16.57±3.20 | **13.90±2.57** | 12.05±0.81 | **9.90±0.69** |
| 18600 | 13.63±2.22 | **10.54±2.45** | 6.80±1.79 | **4.99±1.87** | 15.62±0.85 | **11.61±1.24** | 12.37±0.32 | **10.21±0.96** |

Table 13: Cross-domain effect on single product classifiers trained with reference-guided synthetic images at all scales. Note that here **A**, **B** and **C** stand for 3 products in the SDI dataset while vA, vB, vC and vABC indicate the defects are copied from which reference set.

| Dataset Size | A | | | | |
| --- | --- | --- | --- | --- | --- |
| | Trad-Aug | vA | vB | vC | vABC |
| 4500 | 16.00±1.04 | **12.90±2.61** | 13.08±1.65 | 14.90±2.46 | 15.27±3.49 |
| 6600 | 14.90±1.38 | 13.99±1.89 | **11.26±1.04** | 14.36±4.04 | 16.00±2.85 |
| 12000 | 13.81±2.36 | 11.81±2.65 | 12.72±2.87 | 11.99±1.63 | **11.09±3.49** |
| 18600 | 13.63±2.22 | **12.72±5.22** | 14.36±3.83 | 14.18±5.05 | 13.81±8.56 |

| Dataset Size | B | | | | |
| --- | --- | --- | --- | --- | --- |
| | Trad-Aug | vA | vB | vC | vABC |
| 4500 | 8.79±0.45 | 7.80±2.15 | **5.60±1.14** | 10.19±0.84 | 6.79±1.30 |
| 6600 | 7.60±1.51 | 6.80±1.65 | 7.80±1.10 | 8.00±2.34 | **6.00±1.41** |
| 12000 | 6.80±1.64 | 6.40±1.34 | 6.60±1.52 | 6.59±1.34 | **5.60±1.34** |
| 18600 | 6.80±1.79 | 6.19±1.78 | **4.40±1.14** | 6.60±1.95 | 5.99±1.58 |

| Dataset Size | C | | | | |
| --- | --- | --- | --- | --- | --- |
| | Trad-Aug | vA | vB | vC | vABC |
| 4500 | 17.14±4.62 | 14.85±0.52 | 16.76±2.58 | 13.90±1.98 | **12.00±1.59** |
| 6600 | 15.23±2.33 | 13.14±1.24 | 13.90±2.29 | 14.28±1.34 | **12.57±1.57** |
| 12000 | 16.57±3.20 | 13.14±2.81 | **11.23±0.80** | 14.85±1.73 | 11.42±0.96 |
| 18600 | 15.62±0.85 | 13.71±1.73 | 15.99±6.75 | **12.57±3.26** | 12.95±2.98 |

## E.2 ADDITIONAL FID AND KID RESULTS ON THE SDI DATASET

We provide additional results in the case of training GANs with augmentation methods in Table 14. Augmentation methods like ADA (Karras et al., 2020a) or DiffAug (Zhao et al., 2020) are proposed to adapt GAN training to limited data. We applied these augmentation methods to StyleGAN v2 and BigGAN, because these state-of-art image synthesis methods are not optimized for small dataset. However, incorporating the augmentation methods in training GANs on the SDI dataset is not always beneficial. The performance of StyleGAN v2 is largely degraded when using ADA, potentially due

to the conflict between augmentation methods and the decentralized location of defects—in the SDI dataset, defects can occur anywhere on the surface. This is in contrast to datasets that were used to evaluate the aforementioned augmentation methods in GANs, where the objects are centralized (e.g., ImageNet (Deng et al., 2009), Cifar (Krizhevsky & Hinton, 2009)) and their attributes (e.g., beard, eye glasses in CelebA (Liu et al., 2015)) only occur in specific images parts.

Table 14: Quantitative comparison of DT-GAN with baseline image synthesis methods using FID and KID. Note that the reported values are not comparable between columns, because they are calculated on different training sets. The scores of StarGAN v2 on single products are omitted because generating images with specified background is not possible due to its network design.

| Method | FID↓ | | | | KID↓ | | | |
|---|---|---|---|---|---|---|---|---|
| | A | B | C | All | A | B | C | All |
| Mokady et al. (2020) | 68.69 | 66.90 | 36.21 | 58.63 | 0.050 | 0.036 | 0.030 | 0.036 |
| StarGAN v2 | - | - | - | 37.70 | - | - | - | 0.013 |
| StyleGAN v2 | 90.10 | 52.95 | 138.09 | 35.34 | 0.072 | 0.027 | 0.186 | 0.013 |
| StyleGAN v2 + ADA | 149.66 | 42.75 | 135.69 | 76.16 | 0.138 | 0.019 | 0.191 | 0.055 |
| BigGAN | 235.66 | 192.89 | 193.61 | 151.43 | 0.248 | 0.199 | 0.276 | 0.115 |
| BigGAN + DiffAug | 218.74 | 134.41 | 270.89 | 155.88 | 0.220 | 0.121 | 0.378 | 0.099 |
| Ours | **58.43** | **36.44** | **22.68** | **29.73** | **0.025** | **0.013** | **0.012** | **0.009** |

### E.3 ABLATION STUDY WITH REGARD TO FID AND KID SCORES

We report the FID and KID scores of the ablation study in Table 15. We notice that both subnetworks show positive correlation to each modification except for structural change as in (a) and (e) . Among the two subnetworks, the reference-guided subnetwork outperforms the latent-guided one in the beginning, which is due to the fact that transferring existing contents is easier than generating them from random noise. This effect is also observed in Figure 5. However, the performance of the latent-guided subnetwork improves significantly after applying per-pixel noise injection. The subnetwork can now output non-deterministic foreground contents even for a fixed input vector which results in better visual quality and higher diversity of generated defects. In the main text, the scores of the mixture set are reported.

Table 15: Ablation study with regard to FID and KID scores.

| | FID↓ | | | KID↓ | | |
|---|---|---|---|---|---|---|
| | Latent | Reference | Mix | Latent | Reference | Mix |
| (a) Baseline StarGAN v2 | 37.73 | 37.99 | 37.70 | 0.013 | 0.013 | 0.013 |
| (b) + Style-Content branches | 43.90 | 32.61 | 33.36 | 0.017 | 0.011 | 0.011 |
| (c) + Foreground classifier | 37.14 | **32.34** | **27.69** | 0.014 | 0.011 | **0.008** |
| (d) + Background classifier | 34.12 | 32.50 | 30.23 | 0.011 | 0.011 | 0.010 |
| (e) + Separately decoding foreground and background in $G$ | 48.52 | 38.11 | 34.79 | 0.017 | 0.015 | 0.011 |
| (f) + Anchor foreground domain (e.g. `Normal`) | 43.66 | 37.45 | 32.15 | 0.019 | 0.015 | 0.011 |
| (g) + Noise injection in Mapping Network | **33.05** | 34.42 | 29.73 | **0.009** | **0.011** | 0.009 |

### E.4 ADDITIONAL RESULTS ON THE MVTEC ANOMALY DETECTION DATASET

The MVTec Anomaly Detection dataset (Bergmann et al., 2019) contains 15 different object and texture categories for anomaly detection. The dataset is formed of non-defective image for training and both non-defective and defective images with various kinds of defects for testing. The pixel-level annotations of all defective images are also provided. It is worth noting that the MVTec Anomaly Detection dataset is relatively small scale in number of images, where the number of training images is ranging from 60 to 391. Moreover, the number of defective images for each defect category in the test set is varying only from 8 to 30, which is relatively limited considering the sophisticated pattern of defects.

We conducted image synthesis experiments on a subset of MVTec Anomaly Detection dataset, where we selected four texture categories: **Carpet**, **Leather**, **Wood** and **Tile** for our targeted scenario i.e. surface defects. Furthermore, we aggregated some of the original defect types defined in the MVTec Anomaly Detection dataset into `scratches` and `spots` according to their visual appearance. We then simply added the subset of the MVTec Anomaly Detection dataset to the training set together with the SDI dataset for training DT-GAN. Details of the resulting dataset are shown in Table 17. Note that the small scale of available data posts a major challenge for training generative models.

**Quantitative Evaluation.** We present additional quantitative results on the subset of the MVTec Anomaly Detection dataset in Table 16, following the same evaluation setup as described in Appendix C. As shown in Table 16, our method achieves the best scores in **Carpet** and **Wood**, which supports our claim that DT-GAN generates synthetic images with higher fidelity and more diverse defect. However, we also observe that StyleGAN v2 seems to outperform our method in **Leather** and **Tile**.

Please note that FID and KID are not optimized to evaluate such a small dataset, there the results should only be interpreted together with the qualitative results.

Note the we again omit the FID and KID of StarGAN v2 because it is not cable of generating images for a specified product due to the 'identity-shift', which is also explained in detail in the qualitative evaluation.

Table 16: Quantitative comparison of DT-GAN with baseline image synthesis methods using FID and KID. Note that the reported values are not comparable between columns, because they were calculated on different training sets.

| Method | FID↓ | | | | KID↓ | | | |
|---|---|---|---|---|---|---|---|---|
| | Carpet | Leather | Tile | Wood | Carpet | Leather | Tile | Wood |
| Mokady (2020) | 41.87 | 60.26 | 275.12 | 81.71 | 0.04 | **0.03** | 0.29 | 0.04 |
| StarGAN v2 | - | - | - | - | - | - | - | - |
| StyleGAN v2 | 51.37 | **51.60** | **225.96** | 140.01 | 0.05 | **0.03** | **0.23** | 0.12 |
| BigGAN + DiffAug | 34.47 | 101.70 | 391.54 | 113.32 | 0.03 | 0.07 | 0.42 | 0.07 |
| Ours | **22.79** | 86.13 | 321.35 | **75.83** | **0.01** | 0.07 | 0.36 | **0.03** |

**Qualitative Evaluation.** For qualitative results, we again discuss the 'latent-guided' and 'reference-guided' synthesis separately.

We present the 'latent-guided' image synthesis results of StyleGAN v2 in Figure 11 and Figure 12 and BigGAN in Figure 13 and Figure 14. The results are acquired by training one model for each product and then generating 16 images from randomly sampled latent codes from each of them. As pointed out in Section 4.1.2, both methods can not adapt well on small dataset. They suffer from model collapsing and show signs of overfitting by generating images similar to the training data. For example, StyleGAN v2 generates images either with no clear defect or identical to the training set (e.g., **Leather** in Figure 11 and Product **B** in Figure 12). The overfitting we observe here also explains the better FID and KID scores in Table 16. For **Tile**, we can see clear signs of mode collapse in the generated **Tile** images of StyleGAN v2. Similarly, BigGAN produces images with single mode and abnormal patterns (e.g., grid structure and gray edges). Unlike StyleGAN v2 and BigGAN, StarGAN v2 and our method both require images as input (i.e. **Source**). Therefore, we randomly sampled two `Normal` images and applied eight defects which are generated from randomly sampled latent codes to each of them. As seen in Figure 15 and Figure 16, StarGAN v2 fails to preserve the background from the given input images due to the highly entangled **FG** and **BG**. Also it fails to generates legit and diverse defects without separately modeling the style and the content. In contrast to aforementioned methods, our DT-GAN produces images with higher fidelity and more diversity in defect patterns as shown in Figure 17 and Figure 18. We believe this again prove the importance of style-content separation and **FG/BG** disentanglement, which we introduce in Section 3.1.

For 'reference-guided' image synthesis, the results of Mokady et al. (2020) are shown in Figure 19 and Figure 20 while the results of StarGAN v2 are in Figure 21 and Figure 22. We can observe a

clear shift in color in all the outputs from Mokady et al. (2020). Moreover, Mokady et al. (2020) can only transfer content between **two** domains. In order to perform translation from a non-defective sample to a defective one, we trained a model for each type of defect and for each product. This sums up to be 13 models (`Scratches` and `Spots` for 6 categories and `Scratches` only for **Tile**). The results from the intended use within one background domain can be found on the diagonal and are marked in red in both Figure 19 and Figure 20. We still show the images that we feed in images from other background domains. As expected, the model then fails to preserve the background of given source images and introduce artifacts to the outputs. Similarly, StarGAN v2 does not preserve the background from the input images. Without style-content separation and **FG/BG** disentanglement, we observe that StarGAN v2 encodes the background characteristics together with the foreground content of the reference images, which results in identity-shits in its output images. Moreover, the output images either show no clear defect or contain abnormal patterns which sabotage the fidelity. On the contrary, our method can faithfully transfer the foreground content of reference images across given background of different products as shown in Figure 23 and Figure 24, which demonstrate the effectiveness of the style-content separation and **FG/BG** disentanglement we introduced in Section 3.1.

It is also worth noting that our method can perform cross-domain image synthesis even the desired combination is not presented in the training set. We demonstrate this on product **Tile**, which only has images with `Scratches` but no `Spots`. As shown in Figure 18 and Figure 24, DT-GAN can generated spots one given **Tile** images. However, this kind of transformation is most useful when the desired combination is reasonable for the downstream applications.

**Limitation and Future Work.** We have demonstrated the feasibility of the proposed DT-GAN by incorporating more products from the MVTec Anomaly Detection dataset in our training procedure. Intensive experiments have shown that the generated images from DT-GAN yielded better results compared to the baseline image synthesis methods. However, we noticed that despite the diverse patterns of the generated defects, DT-GAN tends to apply the styles learned from the SDI dataset also to the samples from the MVTec Anomaly Detection dataset. For example, we can observe some "halo" effects in **Leather** and **Wood** in Figure 18 and some of the generated scratches in Figure 17 and Figure 23 are rather weakly pronounced. We hypothesize this can be counteracted by explicitly localizing the defect and enforcing the model to learn conditional relationships between 'styles' and different backgrounds. We aim to address these issues in future work.

Table 17: Overview of our formation of the MVTec Anomaly Detection sub-dataset. The first column represents the original defect types in the MVTec Anomaly Detection dataset while the first row stands for the defect types in our targeted scenario. We list the ID of samples we took from the MVTec Anomaly Detection dataset and show the number of samples in row **Sum**.

(a) Carpet

|        | **Scratches** | **Spots** |
|--------|---------------|-----------|
| Color  | 011, 012, 014, 016, 017 | 000, 003, 004, 007, 015, 018 |
| Thread | 000 - 018 | - |
| Hole   | - | 000 - 016 |
| Sum    | 24 | 23 |

(b) Leather

|        | **Scratches** | **Spots** |
|--------|---------------|-----------|
| Color  | 001, 003, 005, 007, 009, 011, 013, 015, 018 | 000, 002, 006, 008, 010, 012, 014 |
| Cut    | 000 - 018 | - |
| Fold   | 000 - 006, 009 - 016 | - |
| Glue   | 003, 009, 010, 016, 017 | 000 - 002, 005 - 009, 011 - 015, 018 |
| Poke   | - | 000 - 017 |
| Sum    | 48 | 39 |

(c) Tile

|       | **Scratches** | **Spots** |
|-------|---------------|-----------|
| Crack | 000 - 016 | - |
| Sum   | 17 | 0 |

(d) Wood

|          | **Scratches** | **Spots** |
|----------|---------------|-----------|
| Color    | 003, 005 | - |
| Scratch  | 001 - 006, 008 - 010, 013 - 016, 018 - 020 | 000 - 016 |
| Hole     | - | 000 - 004, 006 - 009 |
| Combined | 008 | 001, 002, 009 |
| Sum      | 19 | 12 |

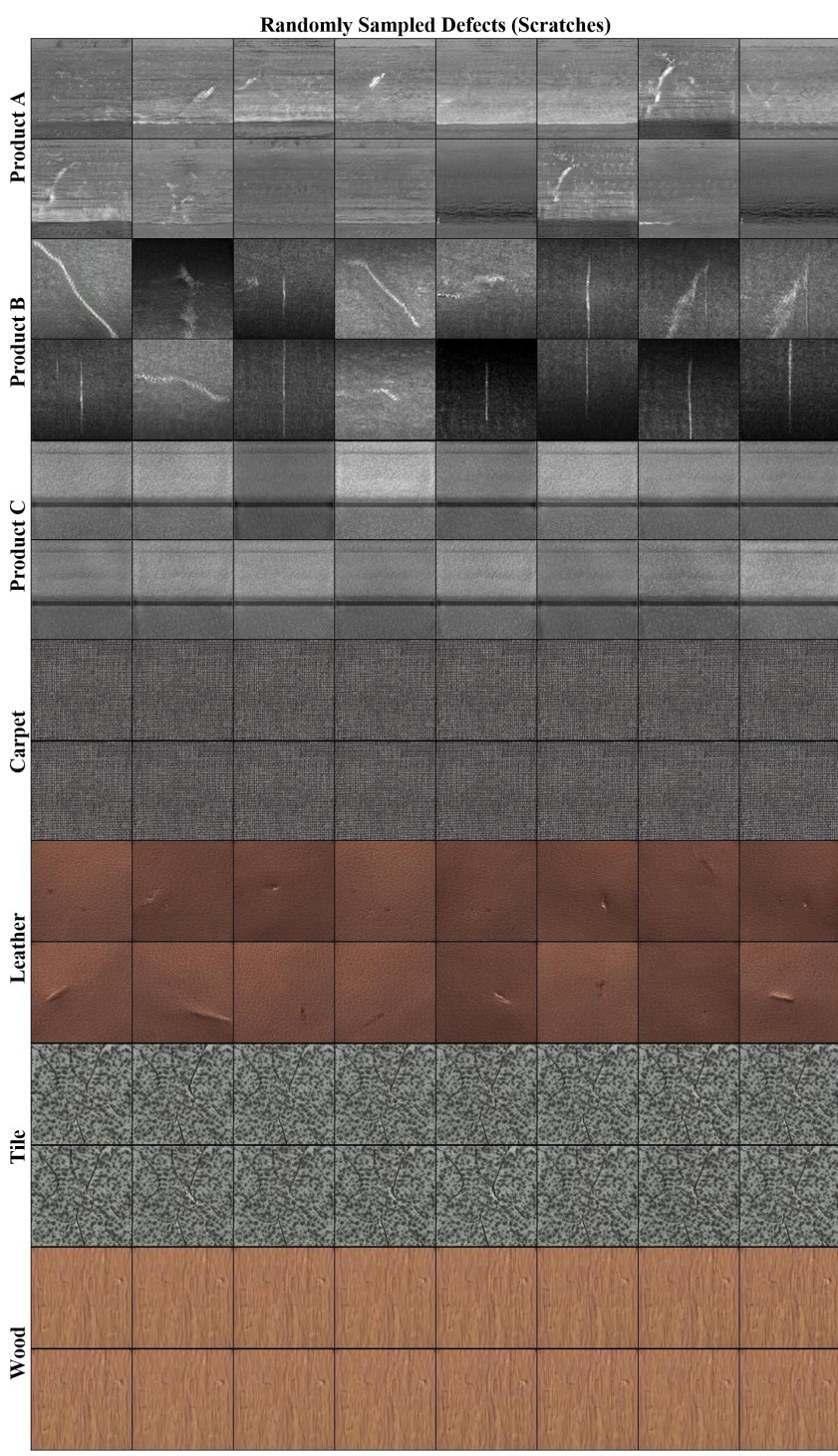

Figure 11: Latent-guided image synthesis results of StyleGAN v2 on the SDI dataset and the MVTec AD dataset. We train a model for each product and generate 16 Scratches images from randomly sampled latent codes.

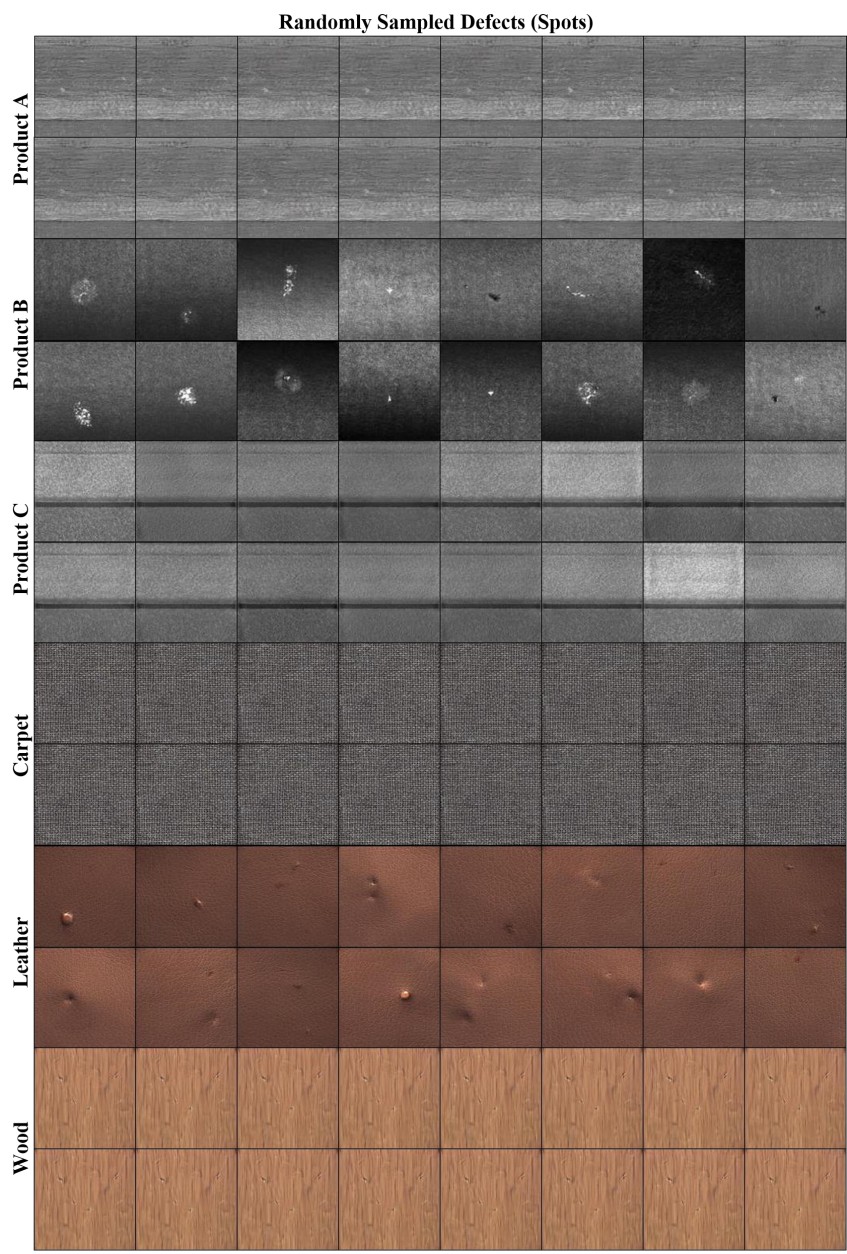

Figure 12: Latent-guided image synthesis results of StyleGAN v2 on the SDI dataset and the MVTec AD dataset. We train a model for each product and generate 16 `Spots` images from randomly sampled latent codes.

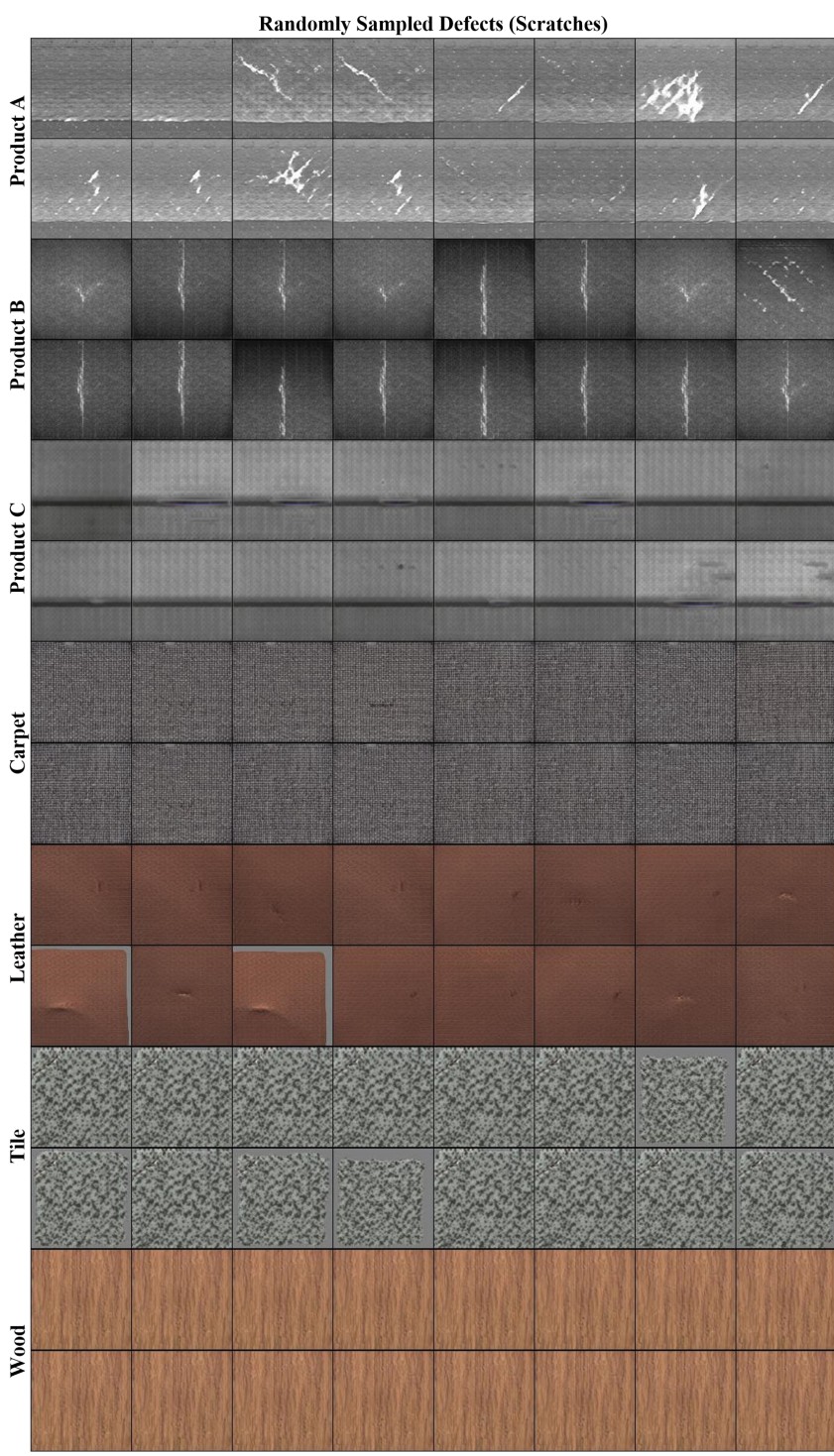

Figure 13: Latent-guided image synthesis results of BigGAN with DiffAug on the SDI dataset and the MVTec AD dataset. We train a model for each product and generate 16 `Scratches` images from randomly sampled latent codes.

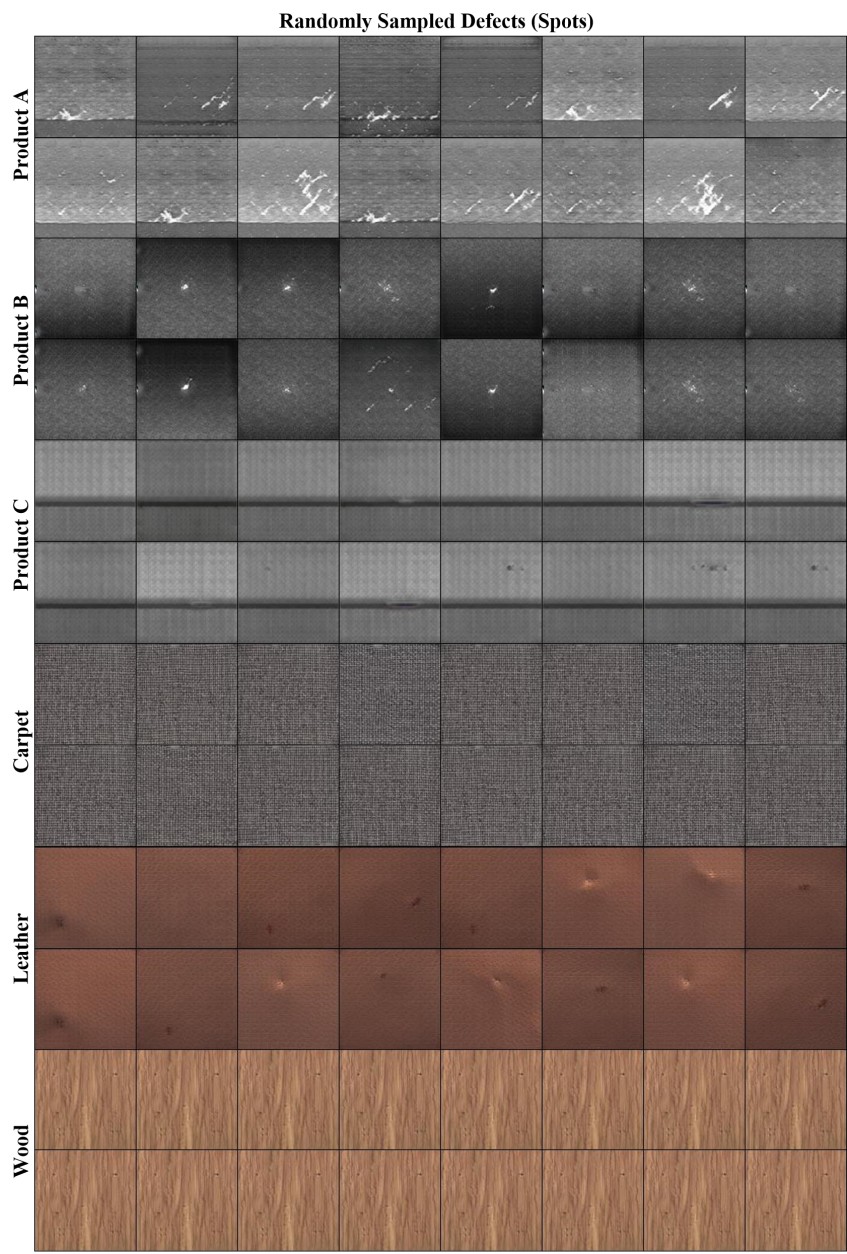

Figure 14: Latent-guided image synthesis results of BigGAN with DiffAug on the SDI dataset and the MVTec AD dataset. We train a model for each product and generate 16 `Spots` images from randomly sampled latent codes.

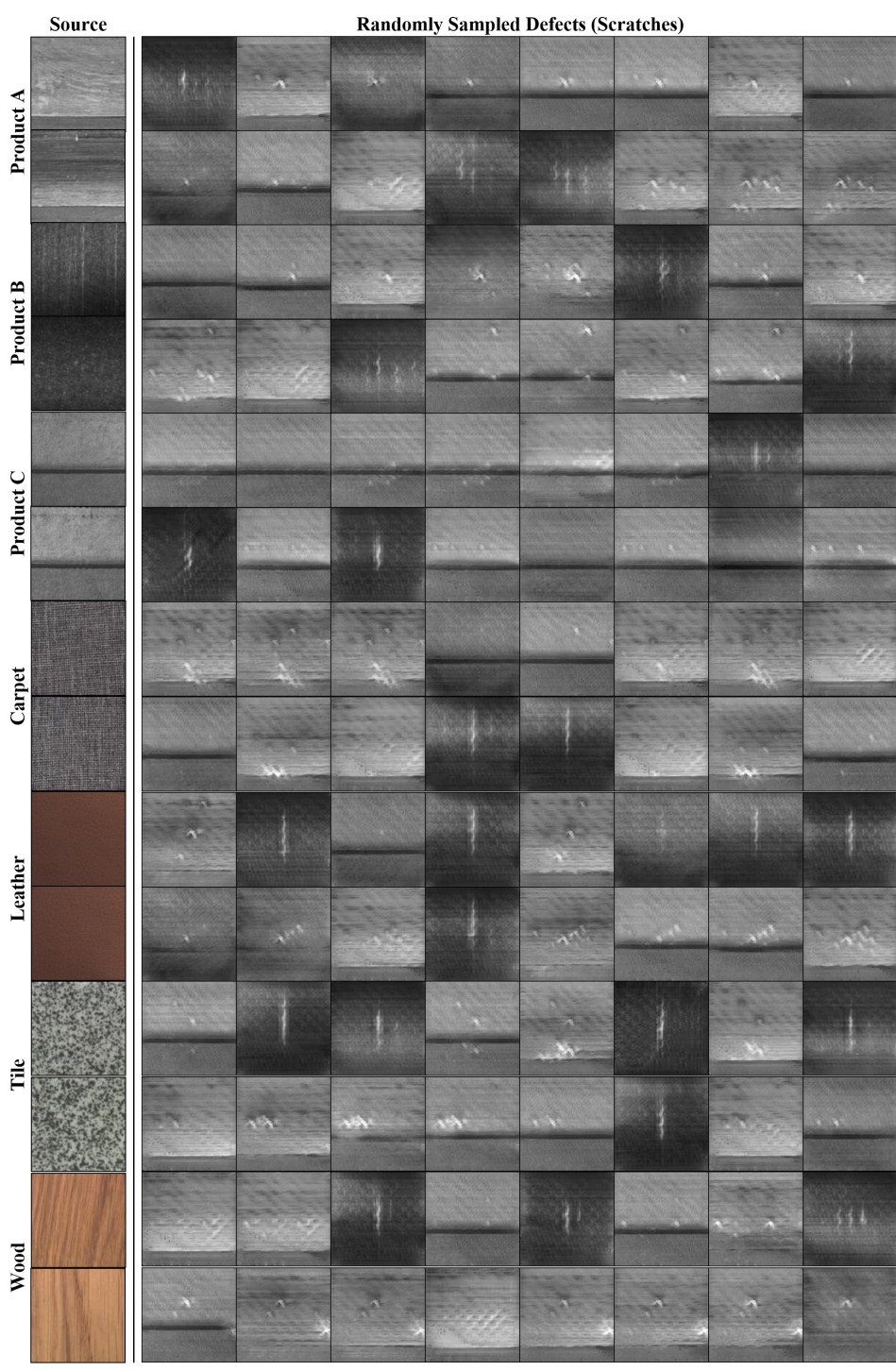

Figure 15: Latent-guided image synthesis results of StarGAN v2 on the SDI dataset and the MVTec AD dataset. The model is trained on a joint set of aforementioned datasets and performs translation from `Normal` to `Scratches`. Note that without the style-content separation and the **FG/BG** disentanglement, StarGAN v2 not only fails to preserve the background from the given **Source** image but also fail to generates legit defects.

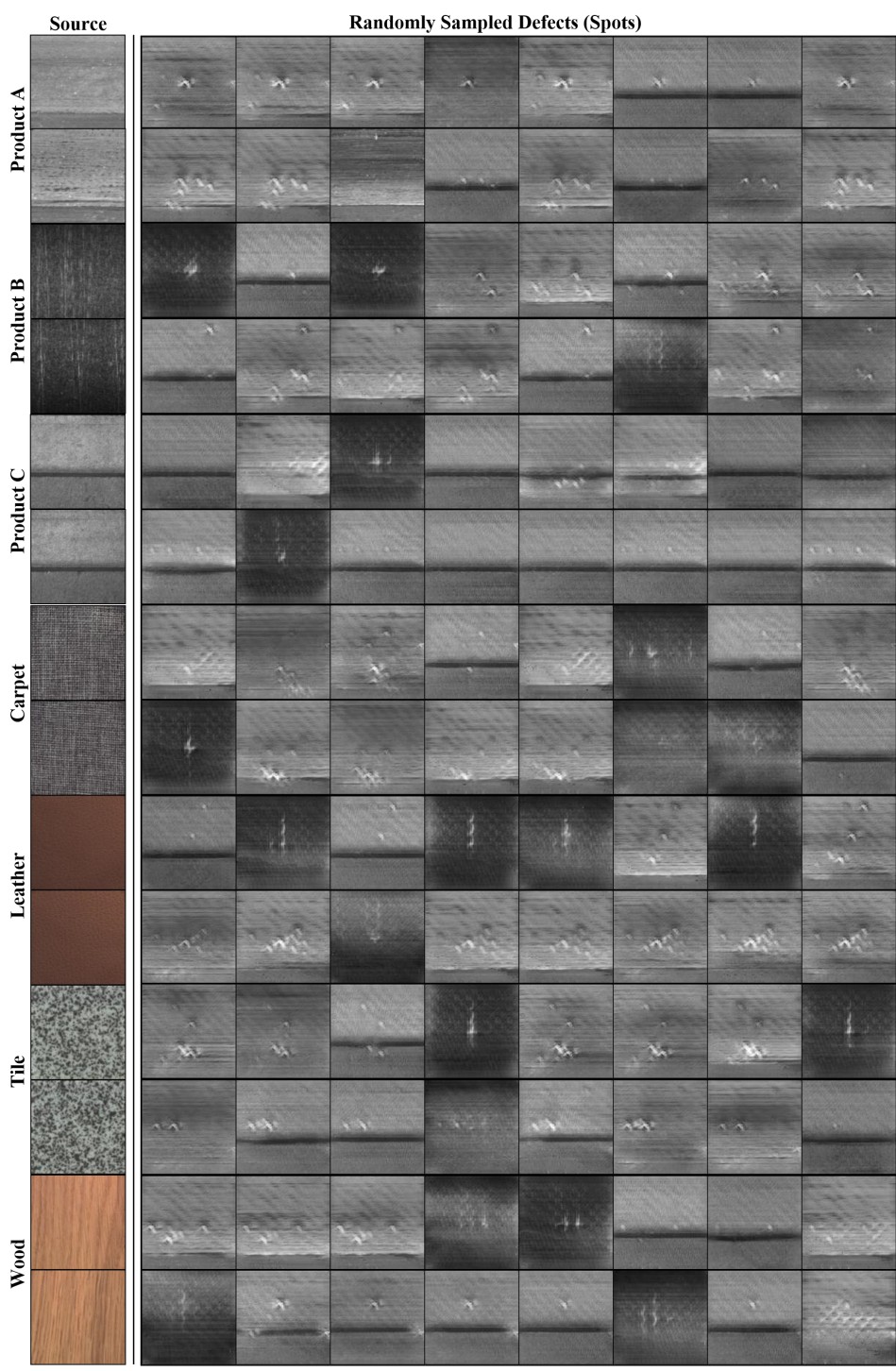

Figure 16: Latent-guided image synthesis results of StarGAN v2 on the SDI dataset and the MVTec AD dataset. The model is trained on a joint set of aforementioned datasets and performs translation from `Normal` to `Spots`. Note that without the style-content separation and the **FG/BG** disentanglement, StarGAN v2 not only fails to preserve the background from the given **Source** image but also fail to generates legit defects.

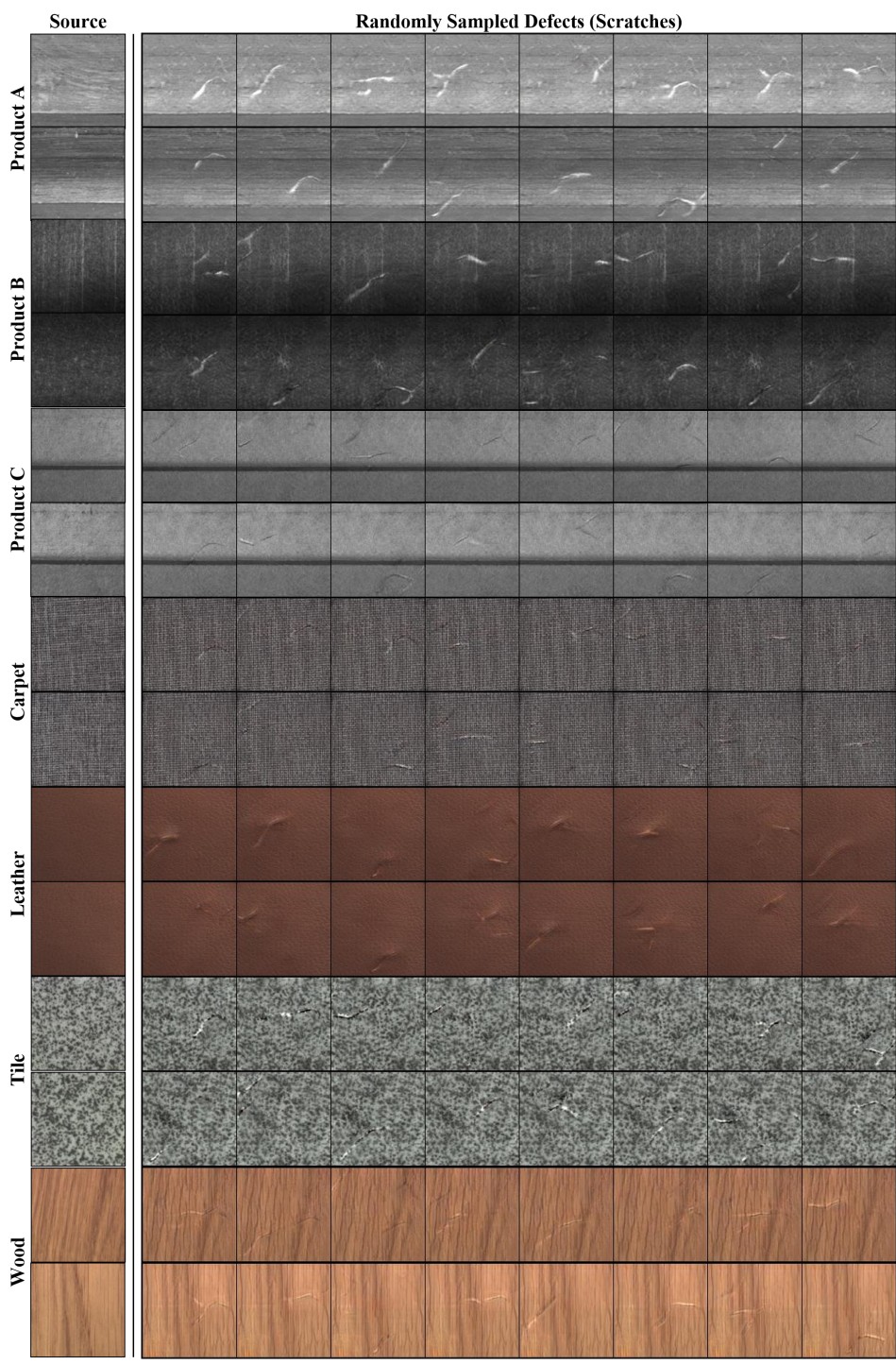

Figure 17: Latent-guided image synthesis results of DT-GAN on the SDI dataset and the MVTec AD dataset. The model is trained on a joint set of aforementioned datasets and performs translation from `Normal` to `Scratches`. Note the our model takes input **Source** images as background and only synthesizes the foreground defects from randomly sampled latent code compared to StyleGAN v2 and BigGAN.

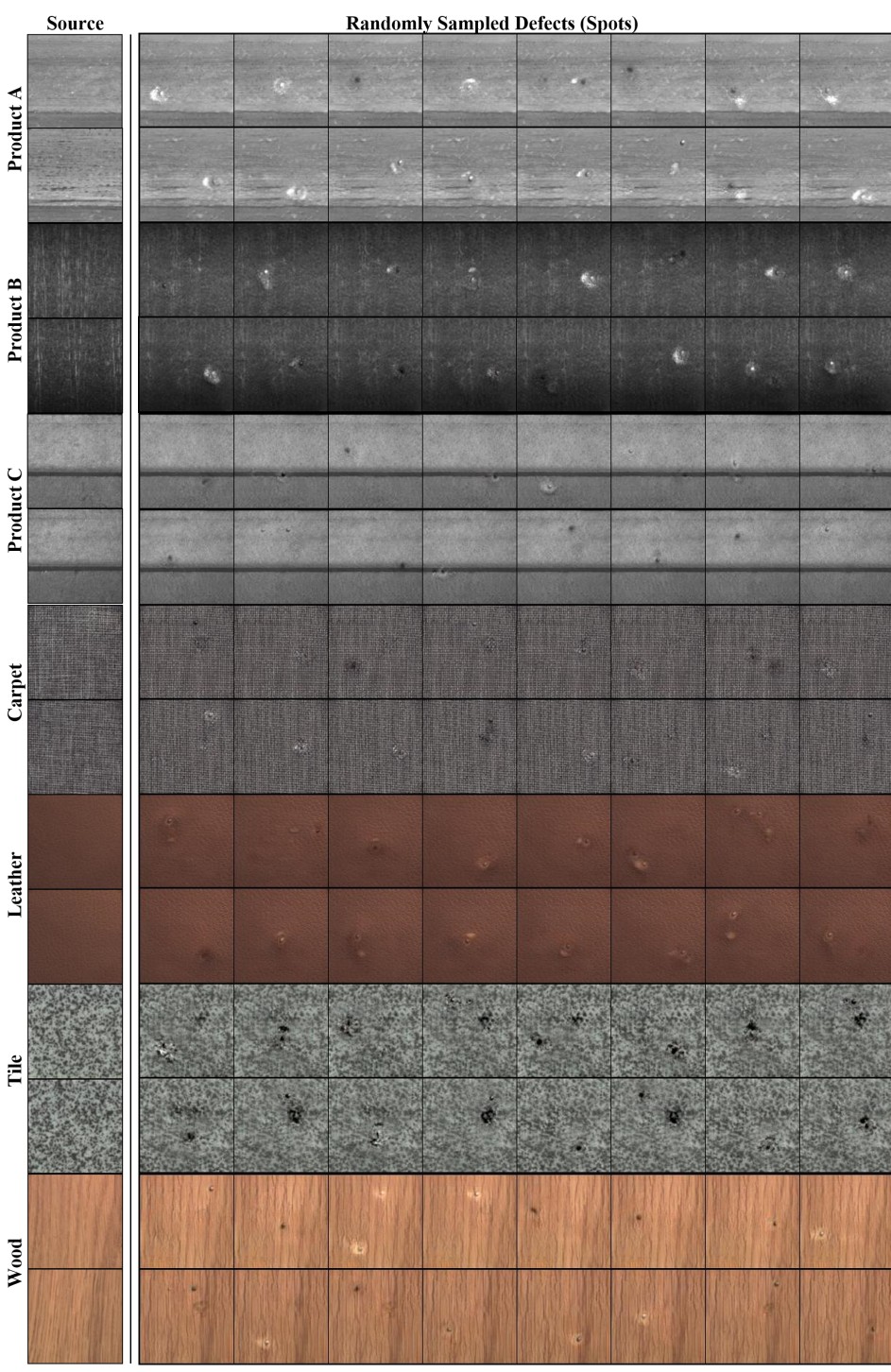

Figure 18: Latent-guided image synthesis results of DT-GAN on the SDI dataset and the MVTec AD dataset. The model is trained on a joint set of aforementioned datasets and performs translation from `Normal` to `Spots`. Note the our model takes input **Source** images as background and only synthesizes the foreground defects from randomly sampled latent code compared to StyleGAN v2 and BigGAN.

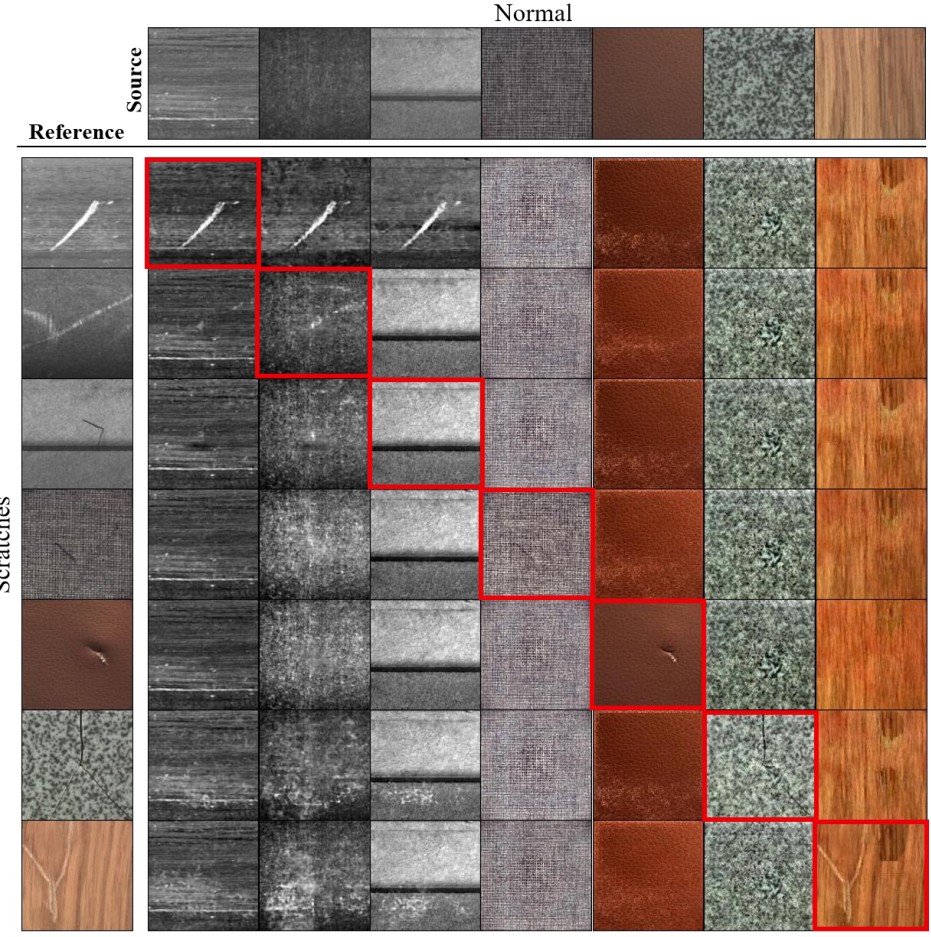

Figure 19: Reference-guided image synthesis results of Mokady et al. (2020) on the SDI dataset and the MVTec AD dataset. We train a model for each product and each defect type. Then we translate `Normal` images to `Scratches` by taking the **Source** as background and applying the foreground defect from **Reference** to it.

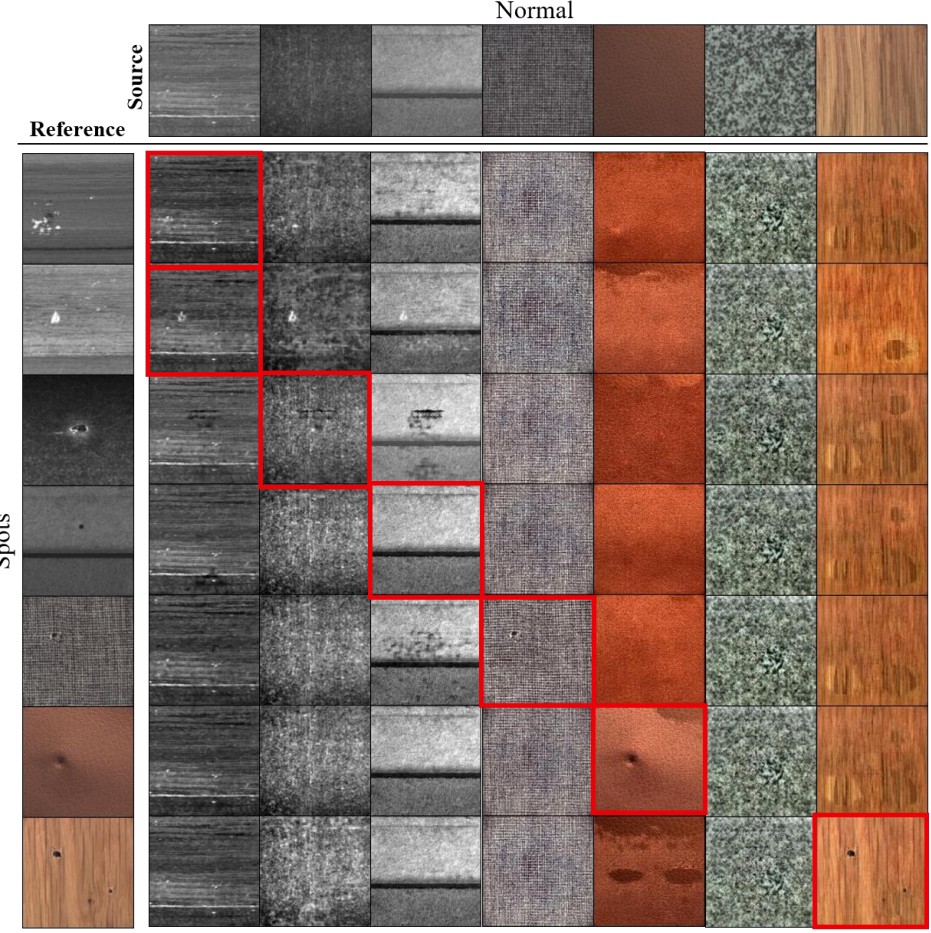

Figure 20: Reference-guided image synthesis results of Mokady et al. (2020) on the SDI dataset and the MVTec AD dataset. We train a model for each product and each defect type. Then we translate `Normal` images to `Spots` by taking the **Source** as background and applying the foreground defect from **Reference** to it.

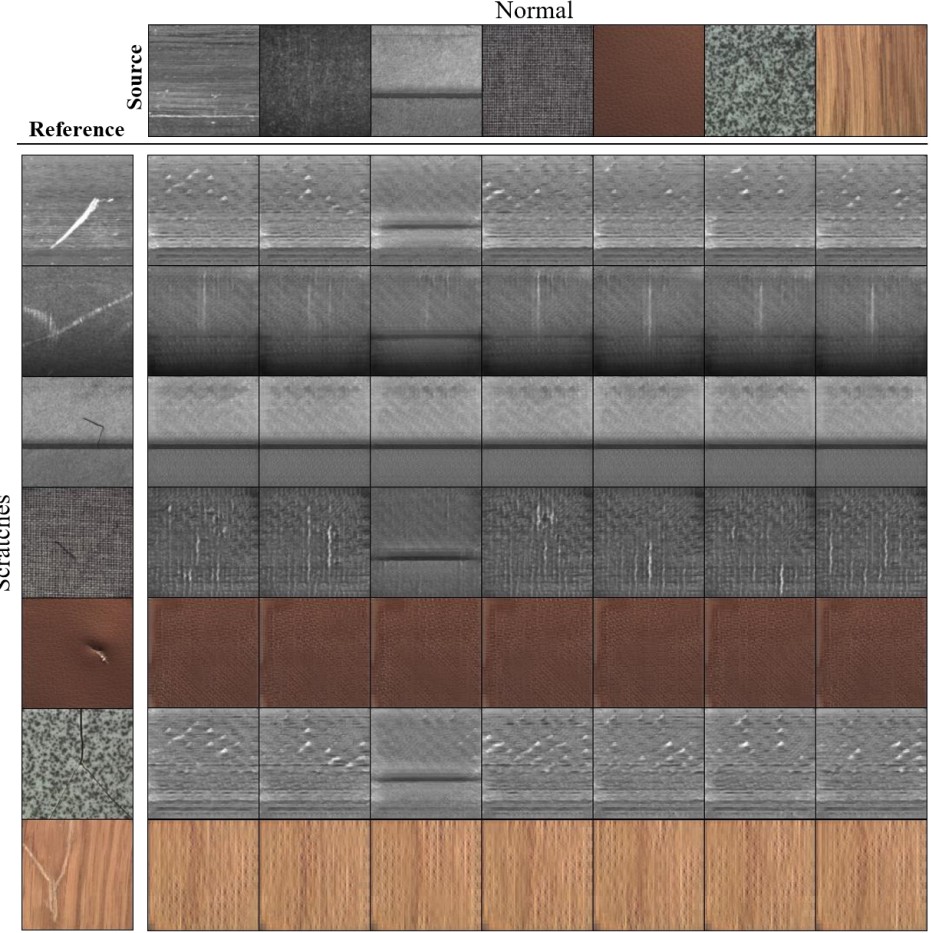

Figure 21: Reference-guided image synthesis results StarGAN v2 on the SDI dataset and the MVTec AD dataset. The model is trained on a joint set of aforementioned datasets and performs translation from `Normal` to `Scratches` by taking the **Source** as background and applying the foreground defect from **Reference** to it. Note that without the style-content separation and the **FG/BG** disentanglement, StarGAN v2 not only fails to preserve the background from the given **Source** image but also fail to generates legit defects.

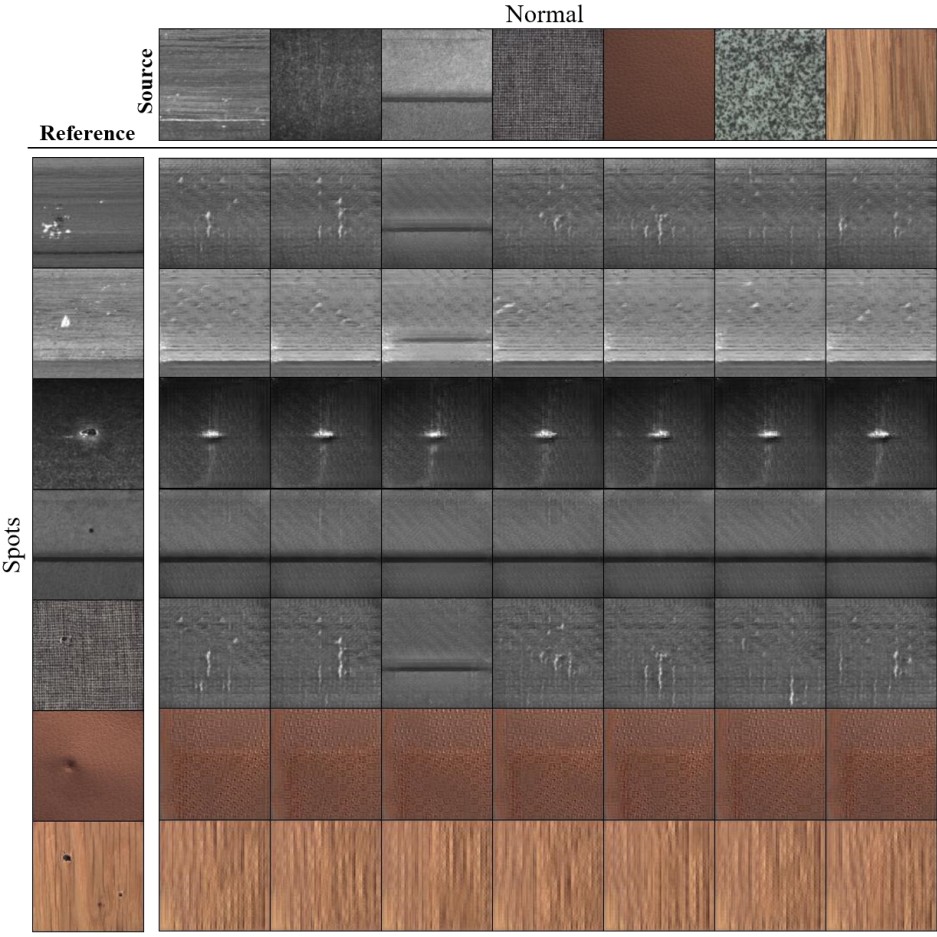

Figure 22: Reference-guided image synthesis results of StarGAN v2 on the SDI dataset and the MVTec AD dataset. The model is trained on a joint set of aforementioned datasets and performs translation from `Normal` to `Spots` by taking the **Source** as background and applying the foreground defect from **Reference** to it. Note that without the style-content separation and the **FG/BG** disentanglement, StarGAN v2 not only fails to preserve the background from the given **Source** image but also fail to generates legit defects.

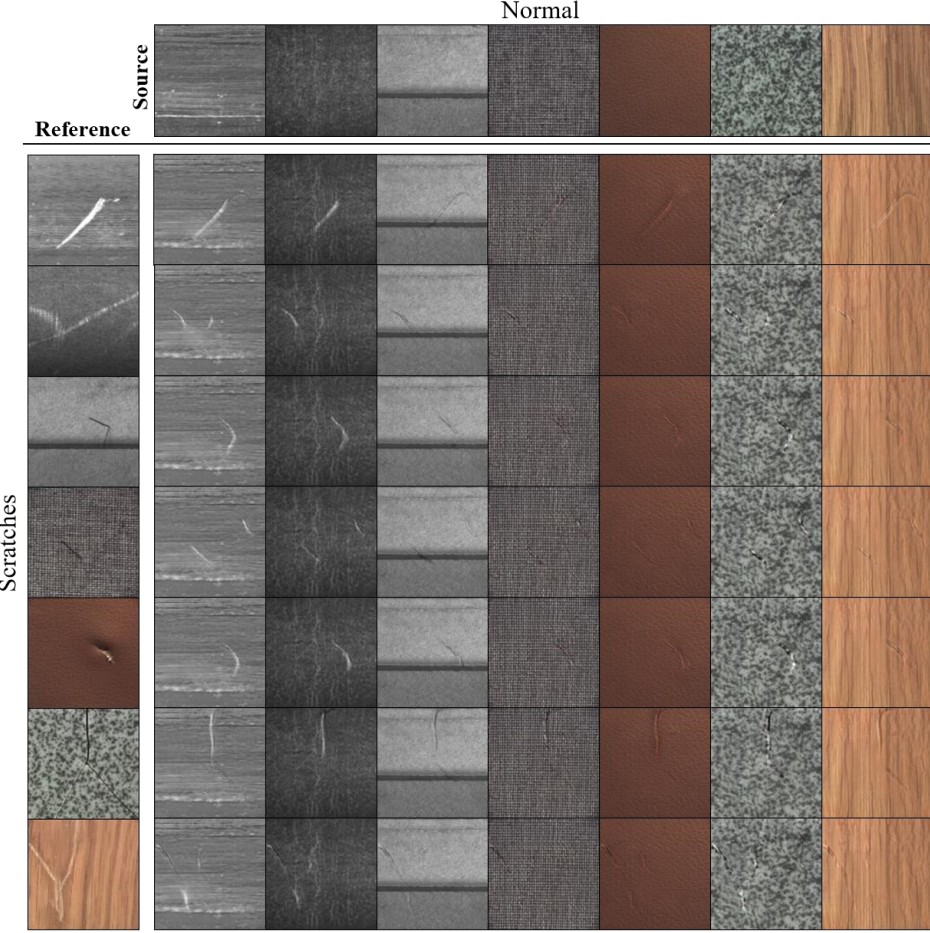

Figure 23: Reference-guided image synthesis results DT-GAN on the SDI dataset and the MVTec AD dataset. The model is trained on a joint set of aforementioned datasets and performs translation from `Normal` to `Scratches` by taking the **Source** as background and applying the foreground defect from **Reference** to it.

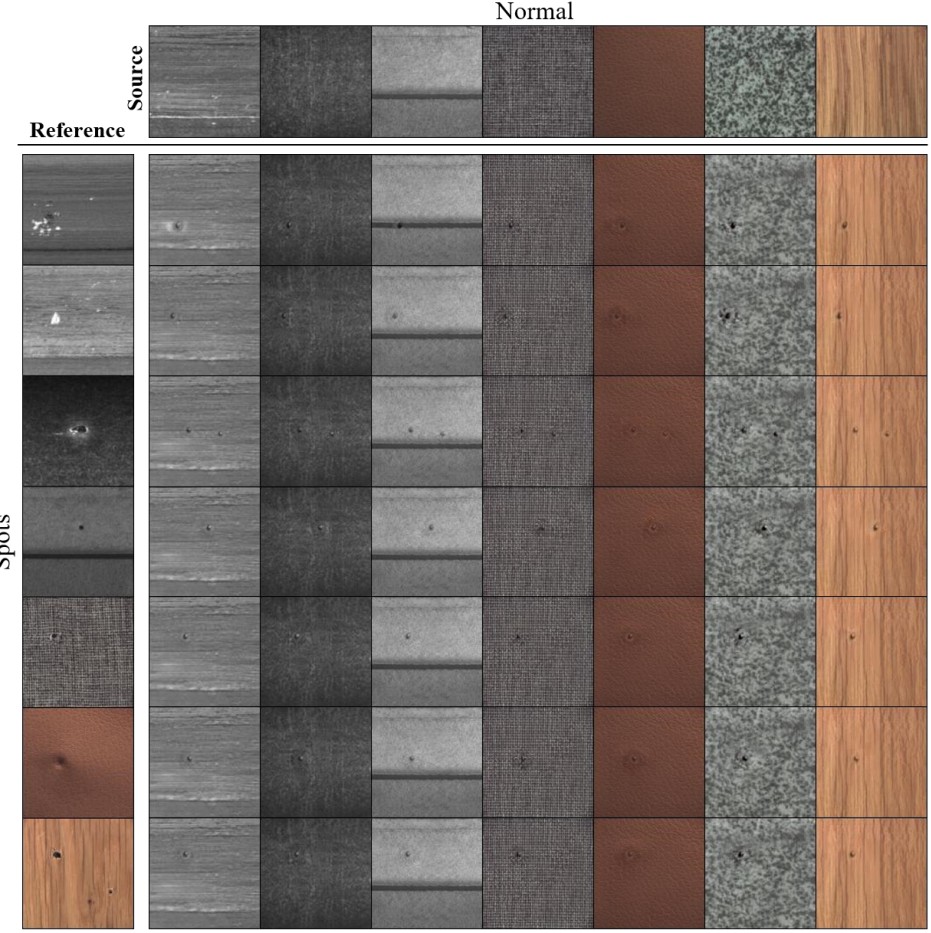

Figure 24: Reference-guided image synthesis results of DT-GAN on the SDI dataset and the MVTec AD dataset. The model is trained on a joint set of aforementioned datasets and performs translation from `Normal` to `Spots` by taking the **Source** as background and applying the foreground defect from **Reference** to it.

