# OpenReview forum: "Defect Transfer GAN: Diverse Defect Synthesis for Data Augmentation"
_ICLR.cc/2022/Conference — ICLR 2022 Submitted_

### Official Review · Reviewer_swLC · 2021-10-25

**Correctness:** 3
**Technical Novelty And Significance:** 2
**Empirical Novelty And Significance:** 2
**Recommendation:** 3
**Confidence:** 5

**Main Review:**

Strength:
The paper is well writen and easy to follow, the proposed methods seems to mitigate the data collection and labelling costs of the defect inspection problem widely available in production industry.

Weakness:
From technical part, the model is basically a tailored StarGAN v2 tuned for the defect generation task, the novelty and contribution of methodology is limited, particularly for ICLR 2022.

The employed SDI dataset is not publicly available and it's difficult for researchers to evaluate and compare the performances.

While StyleGAN v2 and BigGAN are involved for comparison, how do you train them for the synthesis? By fintuning? or training from scratch?

Only scratch and spots are involved, the number of categories for defects are too limited and it's not convincing to support the claim. Much more defects and products need to be tested and evaluated.


**Summary Of The Paper:**


The paper prposes a StarGAN based model to distangle the defect foreground, background, transfer the style of foreground and then synthesize the defected image for different products. The quality of the synthesized defect images are evaluated using FID, KID. The approach is also appled to augment defect images to improve the performance of defect classification.

**Summary Of The Review:**


The work presented in the paper can be regarded as an application of StarGAN v2, while the paper is well writen and easy to follow, the novelty and contribution of methodology is limited, particularly for ICLR 2022.

As only scratch and spots are involved, the number of categories for defects are too limited and it's not convincing to support the claim. Much more defects and products need to be tested and evaluated.

---

> ### Author Response · Authors · 2021-11-12
> **Response to the concern and questions. Part2.**
>
> >2. The employed SDI dataset is not publicly available and it's difficult for researchers to evaluate and compare the performances.
>
> We are planning to make both of the source code and dataset publicly available for further study and evaluation as promised in the reproducibility statement.
>
> >3. While StyleGAN v2 and BigGAN are involved for comparison, how do you train them for the synthesis? By finetuning? or training from scratch?
>
> We trained both of the StyleGAN v2 and BigGAN from scratch because that was also the way we trained our model. All the experiments were conducted on the same amount of data to demonstrate that our model can generate images with higher diversity and more realistic appearance even with limited data from each product. In contrast, StyleGAN v2 and BigGAN suffered from model collapsing and showed signs of overfitting by generating images similar to the training data.
>
> >4. Only scratch and spots are involved, the number of categories for defects are too limited and it's not convincing to support the claim. Much more defects and products need to be tested and evaluated.
>
> We totally agree that the number of products and types of defects are not sufficient to show the strength of our model. We will add additional results on a subset of the MVTec AD dataset in the supplementary material to demonstrate that our model can work with more products by simply adding them to the training dataset. However, we find it hard to acquire suitable datasets to quickly extend the defects categories. For example, the NEU-CLS dataset (https://www.kaggle.com/kaustubhdikshit/neu-surface-defect-database) contains more categories of defects but no non-defective samples. However, it is essential for our model to learn from non-defective samples. We are very happy to take a hint if you are aware of any suitable dataset.
>
> Thank you for your comments!

---

> ### Author Response · Authors · 2021-11-12
> **Response to the concern and questions. Part1.**
>
> >Weakness: 1. From technical part, the model is basically a tailored StarGAN v2 tuned for the defect generation task, the novelty and contribution of methodology is limited, particularly for ICLR 2022.
>
> We apologize if the technical novelty did not become clear in the submission and would like to emphasize the novelties introduced in our work in the following:
>
> - We compute both style and content from a given input instead of only style as in StarGAN v2. It is crucial to separate them as shown in fig. 5 (a) and (b). In fig. 5 (a), the baseline StarGAN v2 cannot differ between style and content, which results in output images that use the background of the reference images (style/content column) instead of the source images. Moreover, without the style/content separation, the style computed by StarGANv2 doesn’t encode the structural patten of the reference images thus the generated defect patterns are simplistic and repetitive (e.g., there is no full scratch in the second row of fig 5(a)).
>
> - We enforce the fg/bg separation by simple but yet effective classification losses (the FG classifier and the BG classifier). Without the classification losses, the model was not constrained to encode the foreground contents into the designed channels at the information bottle neck of the generator. Please compare column (a) to (d) in fig. 5 and one can see that in all rows the correct foreground and background types are generated, but with a shift in brightness which will be addressed in point 3 later.
>
>     The advantage of fg/bg separation is three-fold: first, with the fg/bg separation the Mapping Network and Style-Content Encoder in our model are able to focus on learning the distribution of the foreground contents (e.g., defects). Second, this feature allows our model to collect foreground contents (defects) over different backgrounds (products) to reduce the need for data. In this way, we don’t need to have abundant data for a single product, instead, we can make use of little data from different products that share similar defect patterns. Third, it allows free combination of fg/bg which enriches the diversity of the outputs.
>
>     The models that were trained without fg/bg separation (e.g., StyleGAN v2, BigGAN and StarGAN v2) require more data of a single product for training. For example, StyleGAN v2 and BigGAN suffered from mode collapsing on the SDI dataset. Also, they showed signs of overfitting by generating images similar to the training data. Moreover, without the fg/bg separation, the foreground contents in the generated images are conditioned on the given backgrounds for all three methods. That is, a scratch pattern that present on product A will not be put to product B in the generated images which limits the diversity of the outputs.
>
> - We separately decoded the foreground and background in the generator. Note that the style is only applied to the foreground. With this modification, the model is forced to focus on learning the style of foreground contents (defects) instead of the style of the background. In fig. 5(e), we can see a clear effect after applying this feature: the backgrounds provided by the source images are better preserved than before (with the correct brightness).
>
> - Mokady et al. introduced a concept that the model should be able to identity the difference between ‘two’ domains when one of the domains contains a feature that the other doesn’t have. We refer to this concept as “anchor” and combine it with the multi-task discriminator to work with multiple domains (> 2).  Therefore, our model can not only perform the transformation between non-defective and defective samples, but also between defective samples (see fig. 10 in appendix E).
>
>     The quantitative evaluation of above-mentioned components can be found in Appendix E.
>
> The combination of all these ingredients leads to a model that cannot only transfer existing foreground contents (defects) across different backgrounds (products) but also apply new foreground contents generated from latent codes to the assigned backgrounds. Furthermore, our model can generate images with higher diversity and more realistic appearance when only limited data for each product is available, which StarGAN v2 and other state-of-the-art image synthesis methods could not do before. This is particularly useful when there are only few images with a specific combination of product and defect but plenty of similar defects on other products.

---

> ### Author Response · Authors · 2021-11-22
> **Thank you for the advice, we revised the paper with regard to your suggestions.**
>
> >Weakness:
> >1. From technical part, the model is basically a tailored StarGAN v2 tuned for the defect generation task, the novelty and contribution of methodology is limited, particularly for ICLR 2022.
>
> We apologize if the technical novelty did not become clear in the first submission. We now revised Section 3 in the main paper and emphasized more on our novelties.
>
> >4. Only scratch and spots are involved, the number of categories for defects are too limited and it's not convincing to support the claim. Much more defects and products need to be tested and evaluated.
>
> In the revised version, please find the experiment results on the MVTec AD Dataset in Appendix E.4.

---

### Official Review · Reviewer_Ks5k · 2021-11-01

**Correctness:** 3
**Technical Novelty And Significance:** 2
**Empirical Novelty And Significance:** 3
**Recommendation:** 3
**Confidence:** 4

**Main Review:**

**Strengths**

- Applying GANs to defect synthesis is interesting.

- The results of reference-guided defect transfer are promising.

**Weaknesses**

- My main concern is the technical novelty. From the task perspective, using GANs for data augmentation is not new, and using GANs for style/content transfer is also not new. This paper just applies it to a new area. From the technique perspective, most of the architectures are borrowed from StarGAN v2. Some loss terms are added to improve the performance, but all losses seem to be trivial. Despite the good results, I cannot see many insights from this work (*e.g.*, how can this work guide other researchers?)

- The proposed approach is only evaluated on the Surface Defect Inspection (SDI) dataset. The superiority cannot be fully verified.

**Summary Of The Paper:**

This work applies GAN for data augmentation to enhance defect classification. Concretely, based on StarGAN v2 structure, the proposed framework is able to encode the foreground and the background separately, enabling diverse defect synthesis by either transferring the defect from a reference sample or synthesizing from randomly sampled noises.

**Summary Of The Review:**

To be honest, I do not think this work is suitable for ICLR as most audiences may not get interested. I strongly recommend the authors submit this work to a conference/journal in the industry field.

---

> ### Author Response · Authors · 2021-11-12
> **Response to the concern and questions. Part2.**
>
> >-   The proposed approach is only evaluated on the Surface Defect Inspection (SDI) dataset. The superiority cannot be fully verified.
>
> We are planning to make both of the source code and dataset publicly available for further study and evaluation as promised in the reproducibility statement.
>
> Additional results on a subset of the MVTec AD dataset will be added to the supplementary material to demonstrate that our model can work with more products by simply adding them to the training dataset.
>
> Thank you for your comments!

---

> ### Author Response · Authors · 2021-11-12
> **Response to the concern and questions. Part1.**
>
> >-   My main concern is the technical novelty. From the task perspective, using GANs for data augmentation is not new, and using GANs for style/content transfer is also not new. This paper just applies it to a new area. From the technique perspective, most of the architectures are borrowed from StarGAN v2. Some loss terms are added to improve the performance, but all losses seem to be trivial. Despite the good results, I cannot see many insights from this work (_e.g._, how can this work guide other researchers?)
>
> We apologize if the technical novelty did not become clear in the submission and would like to emphasize the novelties introduced in our work in the following:
>
> - We compute both style and content from a given input instead of only style as in StarGAN v2. It is crucial to separate them as shown in fig. 5 (a) and (b). In fig. 5 (a), the baseline StarGAN v2 cannot differ between style and content, which results in output images that use the background of the reference images (style/content column) instead of the source images. Moreover, without the style/content separation, the style computed by StarGANv2 doesn’t encode the structural patten of the reference images thus the generated defect patterns are simplistic and repetitive (e.g., there is no full scratch in the second row of fig 5(a)).
>
> - We enforce the fg/bg separation by simple but yet effective classification losses (the FG classifier and the BG classifier). Without the classification losses, the model was not constrained to encode the foreground contents into the designed channels at the information bottle neck of the generator. Please compare column (a) to (d) in fig. 5 and one can see that in all rows the correct foreground and background types are generated, but with a shift in brightness which will be addressed in point 3 later.
>
>     The advantage of fg/bg separation is three-fold: first, with the fg/bg separation the Mapping Network and Style-Content Encoder in our model are able to focus on learning the distribution of the foreground contents (e.g., defects). Second, this feature allows our model to collect foreground contents (defects) over different backgrounds (products) to reduce the need for data. In this way, we don’t need to have abundant data for a single product, instead, we can make use of little data from different products that share similar defect patterns. Third, it allows free combination of fg/bg which enriches the diversity of the outputs.
>
>     The models that were trained without fg/bg separation (e.g., StyleGAN v2, BigGAN and StarGAN v2) require more data of a single product for training. For example, StyleGAN v2 and BigGAN suffered from mode collapsing on the SDI dataset. Also, they showed signs of overfitting by generating images similar to the training data. Moreover, without the fg/bg separation, the foreground contents in the generated images are conditioned on the given backgrounds for all three methods. That is, a scratch pattern that present on product A will not be put to product B in the generated images which limits the diversity of the outputs.
>
> - We separately decoded the foreground and background in the generator. Note that the style is only applied to the foreground. With this modification, the model is forced to focus on learning the style of foreground contents (defects) instead of the style of the background. In fig. 5(e), we can see a clear effect after applying this feature: the backgrounds provided by the source images are better preserved than before (with the correct brightness).
>
> - Mokady et al. introduced a concept that the model should be able to identity the difference between ‘two’ domains when one of the domains contains a feature that the other doesn’t have. We refer to this concept as “anchor” and combine it with the multi-task discriminator to work with multiple domains (> 2).  Therefore, our model can not only perform the transformation between non-defective and defective samples, but also between defective samples (see fig. 10 in appendix E).
>
>     The quantitative evaluation of above-mentioned components can be found in Appendix E.
>
> The combination of all these ingredients leads to a model that cannot only transfer existing foreground contents (defects) across different backgrounds (products) but also apply new foreground contents generated from latent codes to the assigned backgrounds. Furthermore, our model can generate images with higher diversity and more realistic appearance when only limited data for each product is available, which StarGAN v2 and other state-of-the-art image synthesis methods could not do before. This is particularly useful when there are only few images with a specific combination of product and defect but plenty of similar defects on other products.

---

> ### Author Response · Authors · 2021-11-22
> **Thank you for the advice, we revised the paper with regard to your suggestions.**
>
>
> >-   My main concern is the technical novelty. From the task perspective, using GANs for data augmentation is not new, and using GANs for style/content transfer is also not new. This paper just applies it to a new area. From the technique perspective, most of the architectures are borrowed from StarGAN v2. Some loss terms are added to improve the performance, but all losses seem to be trivial. Despite the good results, I cannot see many insights from this work (_e.g._, how can this work guide other researchers?)
>
> We apologize if the technical novelty did not become clear in the first submission. We now revised Section 3 in the main paper and emphasized more on our novelties.
>
> >-   The proposed approach is only evaluated on the Surface Defect Inspection (SDI) dataset. The superiority cannot be fully verified.
>
> In the revised version, please find the experiment results on the MVTec AD Dataset in Appendix E.4.

---

> > ### Comment · Reviewer_Ks5k · 2021-11-30
> > **Thanks for the revision**
> >
> > I appreciate the authors' effort in the response and the revision. But I agree with other reviewers about the limited technical contribution of this work. Again, I strongly recommend the authors submit this work to a conference/journal in the industry field.

---

### Official Review · Reviewer_NBex · 2021-11-02

**Correctness:** 2
**Technical Novelty And Significance:** 2
**Empirical Novelty And Significance:** 2
**Recommendation:** 5
**Confidence:** 4

**Main Review:**

> It is hard to understand what is the style and what is the content in the targeted application. Even though the method is proposed to tackle the defect recognition problem; while style and content are not clearly defined in such a context. This confuses the concept of style and content throughout the draft.

> The technical novelty looks weak: Most of components are similar to the StarGAN v2. While authors insist that they added few more modules such as background/foreground classifier, however this extension is rather trivial. Method-wise it is hard to find the improvement. In the application-side there could be, however it was not clearly explained why such style/content separation and bg/fg classification are important for the targeted application.

> Experiments are somehow limited: Only 1 dataset is involved for the experiment. Furthermore, the presentation for the results is rather weak. Figures 3 through 6 are not clearly explained. They are not the natural images and it is hard to empathize that the images obtained is operating well for the given scenario. I think authors need to better explain such a subtle improvement.

**Summary Of The Paper:**

Authors proposed the GAN method that can translate the defect between images. The obtained results from the proposed GAN is able to disentangle the defect-specific content from the background images in the weakly-supervised manner. By using the proposed GAN method, authors augmented the training data. The accuracy gain obtained using the data augmentation shows consistent and significant gap to the conventional method without the data augmentation.

**Summary Of The Review:**

Due to the limited presentation, experiments and technical novelty, I am on the borderline for this draft yet. However, I could go towards the accept, if authors could made effective rebuttal to my comments.

---

> ### Author Response · Authors · 2021-11-12
> **Response to the concern and questions. Part3.**
>
> >3. Experiments are somehow limited: Only 1 dataset is involved for the experiment. Furthermore, the presentation for the results is rather weak. Fig.s 3 through 6 are not clearly explained. They are not the natural images and it is hard to empathize that the images obtained is operating well for the given scenario. I think authors need to better explain such a subtle improvement.
>
> We will update the supplementary material with additional results on a subset of the MVTec AD dataset which demonstrate that our model can work with more products by simply adding them to the training dataset.
>
> Furthermore, we will improve the presentation for the results to be clearer about what to check and what to expect. Here we would like to first give a quick overview:
>
> -  In fig 3, we compare the ability of generating defective images from randomly sampled latent codes using baselines from related works. In this case, both StyleGAN v2 and BigGAN generate synthetic images purely from randomly sampled latent code while our model only spends model capacity on generating foreground contents and fuses them with given input backgrounds.
>
>     Moreover, the fg/bg separation in our design allows collecting foreground contents (defects) over different backgrounds (products) to learn the distribution of defect patterns. This allows our model to generate defects with higher diversity and more natural appearance when only limited defective samples of each product are available. Also see the improved FID and KID scores in table 1.
>
>     In contrast, StyleGAN v2 often failed to produce defects as seen in the third row of fig. 3(a), also the first and third row in fig. 3(b). On the other hand, images generated by BigGAN seemed to be legit, however, it produced undesired grid patterns in all generated images, which can be observed in fig. 3. Besides, we also observed that defects produced by StyleGAN v2 and BigGAN resemble the few training samples quite closely when they were trained on a small dataset.
>
>     The third row in fig 3(a) and all three rows in fig 3(b) of StarGAN v2 demonstrate the importance of style/content and fg/bg separation: as seen in the fig., StarGAN v2 often failed to preserve the input background, while our model not only preserved the characteristic of the input background but also generated legit defects on it.
>
> - In fig 4, we demonstrate that our model can transfer foreground contents (defects) from a reference image onto a given background. We compare our results with Mokady et al. and stargan v2. As seen in fig 4, Mokady et al. either failed to transfer the contents (defects) to a given background or it simply transferred the whole reference image to the output. StarGAN v2 showed similar failure patterns. On the contrary, our model produced synthetic images that fuse the defects presented in the reference images and the given backgrounds.
>
> Please find the explanation of fig. 5 and 6 in point 1 and 2 above.
>
> Thank you for your comments!

---

> ### Author Response · Authors · 2021-11-12
> **Response to the concern and questions. Part2.**
>
> >2. The technical novelty looks weak: Most of components are similar to the StarGAN v2. While authors insist that they added few more modules such as background/foreground classifier, however this extension is rather trivial. Method-wise it is hard to find the improvement. In the application-side there could be, however it was not clearly explained why such style/content separation and bg/fg classification are important for the targeted application.
>
> We apologize if the technical novelty did not become clear in the submission and would like to emphasize the novelties introduced in our work in the following:
>
> - We compute both style and content from a given input instead of only style as in StarGAN v2. It is crucial to separate them as shown in fig. 5 (a) and (b). In fig. 5 (a), the baseline StarGAN v2 cannot differ between style and content, which results in output images that use the background of the reference images (style/content column) instead of the source images. Moreover, without the style/content separation, the style computed by StarGANv2 doesn’t encode the structural patten of the reference images thus the generated defect patterns are simplistic and repetitive (e.g., there is no full scratch in the second row of fig 5(a)).
>
> - We enforce the fg/bg separation by simple but yet effective classification losses (the FG classifier and the BG classifier). Without the classification losses, the model was not constrained to encode the foreground contents into the designed channels at the information bottle neck of the generator. Please compare column (a) to (d) in fig. 5 and one can see that in all rows the correct foreground and background types are generated, but with a shift in brightness which will be addressed in point 3 later.
>
>     The advantage of fg/bg separation is three-fold: first, with the fg/bg separation the Mapping Network and Style-Content Encoder in our model are able to focus on learning the distribution of the foreground contents (e.g., defects). Second, this feature allows our model to collect foreground contents (defects) over different backgrounds (products) to reduce the need for data. In this way, we don’t need to have abundant data for a single product, instead, we can make use of little data from different products that share similar defect patterns. Third, it allows free combination of fg/bg which enriches the diversity of the outputs.
>
>     The models that were trained without fg/bg separation (e.g., StyleGAN v2, BigGAN and StarGAN v2) require more data of a single product for training. For example, StyleGAN v2 and BigGAN suffered from mode collapsing on the SDI dataset. Also, they showed signs of overfitting by generating images similar to the training data. Moreover, without the fg/bg separation, the foreground contents in the generated images are conditioned on the given backgrounds for all three methods. That is, a scratch pattern that present on product A will not be put to product B in the generated images which limits the diversity of the outputs.
>
> - We separately decoded the foreground and background in the generator. Note that the style is only applied to the foreground. With this modification, the model is forced to focus on learning the style of foreground contents (defects) instead of the style of the background. In fig. 5(e), we can see a clear effect after applying this feature: the backgrounds provided by the source images are better preserved than before (with the correct brightness).
>
> - Mokady et al. introduced a concept that the model should be able to identity the difference between ‘two’ domains when one of the domains contains a feature that the other doesn’t have. We refer to this concept as “anchor” and combine it with the multi-task discriminator to work with multiple domains (> 2).  Therefore, our model can not only perform the transformation between non-defective and defective samples, but also between defective samples (see fig. 10 in appendix E).
>
>     The quantitative evaluation of above-mentioned components can be found in Appendix E.
>
> The combination of all these ingredients leads to a model that cannot only transfer existing foreground contents (defects) across different backgrounds (products) but also apply new foreground contents generated from latent codes to the assigned backgrounds. Furthermore, our model can generate images with higher diversity and more realistic appearance when only limited data for each product is available, which StarGAN v2 and other state-of-the-art image synthesis methods could not do before. This is particularly useful when there are only few images with a specific combination of product and defect but plenty of similar defects on other products.

---

> ### Author Response · Authors · 2021-11-12
> **Response to the concern and questions. Part1.**
>
> >1. It is hard to understand what is the style and what is the content in the targeted application. Even though the method is proposed to tackle the defect recognition problem; while style and content are not clearly defined in such a context. This confuses the concept of style and content throughout the draft.
>
> In our targeted application, the content represents the shape and location of defect (similar to a saliency map) while the style of defect represents the artistic look such as heavy or light stroke of a scratch. We demonstrate the effect of applying different styles on the same defect content in fig. 6.
>
> Given a fixed pair of background and foreground (defect), our model is cable of generating different outputs according to different style code (see column 3-5 in fig. 6 (a) and (b)).
>
> We argue that while the content (defect) is the main target to recognize in our target application, different styles applied to the content increase the diversity of the generated images.

---

> ### Author Response · Authors · 2021-11-22
> **Thank you for the advice, we revised the paper with regard to your suggestions.**
>
>
> >1.  It is hard to understand what is the style and what is the content in the targeted application. Even though the method is proposed to tackle the defect recognition problem; while style and content are not clearly defined in such a context. This confuses the concept of style and content throughout the draft.
> >2.  The technical novelty looks weak: Most of components are similar to the StarGAN v2. While authors insist that they added few more modules such as background/foreground classifier, however this extension is rather trivial. Method-wise it is hard to find the improvement. In the application-side there could be, however it was not clearly explained why such style/content separation and bg/fg classification are important for the targeted application.
>
> We apologize if the technical novelty did not become clear in the first submission. We now revised Section 3 in the main paper and emphasized more on our novelties.
>
> >3. Experiments are somehow limited: Only 1 dataset is involved for the experiment. Furthermore, the presentation for the results is rather weak. Fig.s 3 through 6 are not clearly explained. They are not the natural images and it is hard to empathize that the images obtained is operating well for the given scenario. I think authors need to better explain such a subtle improvement.
>
> In the revised version, please find the experiment results on the MVTec AD Dataset in Appendix E.4.

---

> > ### Comment · Reviewer_NBex · 2021-11-29
> > **Thank you for the response.**
> >
> > I still think the novelty of the paper is rather weak even though minor changes have been made in the draft. At this stage, I 'd like to remain in the same same score, 'marginally below the threshold'.

---

### Official Review · Reviewer_GYiR · 2021-11-03

**Correctness:** 3
**Technical Novelty And Significance:** 2
**Empirical Novelty And Significance:** 2
**Recommendation:** 3
**Confidence:** 4

**Main Review:**

Strength:
+ The paper addresses an interesting and important application.

+ The technical implementation details are well documented. The appendix provides great details on the experimental protocol and additional results. The authors promise to publish the code and dataset so this could be a nice contribution to the community.

+ The classification accuracy was higher when using the proposed data synthesis approach.

Weakness:
-	I have a hard time to find anything that is technically novel. All the training losses are inherent from StartGANv2 or CycleGAN. The main difference lies in the foreground/background separation, but there was not ablation study to validate this.

-	Only results on a *closed* dataset is reported. I feel that this paper would be a lot more better suited for conferences with a focus on applications. I am not sure what’s the take-away lesson on the technical components from this paper.

**Summary Of The Paper:**

 This paper introduces Defect Transfer GAN to transfer or generate different types of defects such as stretches or spots (foreground) and apply them onto different product images (background). The method builds upon StarGANv2 and add the cycle/content consistency loss and classification loss between foreground and background. Experiments were reported on the Surface Defect Inspection dataset. Results show that the proposed method achieved better FID and KID scores. The paper shows that training on the synthesized dataset performs better than traditional data augmentation.

**Summary Of The Review:**

I think this paper tackles an interesting applications. The results are thorough and solid. My main concern of this paper is that the technical novelty is limited. I am thus leaning negative about this paper.

---

> ### Author Response · Authors · 2021-11-12
> **Response to the concern and questions. Part2.**
>
> >-   Only results on a _closed_ dataset is reported. I feel that this paper would be a lot more better suited for conferences with a focus on applications. I am not sure what’s the take-away lesson on the technical components from this paper.
>
> Additional results from experiments conducted on a subset of the MVTec AD dataset will be added to the supplementary material to show our model can adapt to more products than the three products in the SDI dataset.
>
> We will revise the conclusion to convey a clearer take-away lesson.
>
> Thank you for your comments!

---

> > ### Comment · Reviewer_GYiR · 2021-11-30
> > **Thank you for the response and revision**
> >
> > I appreciate the authors' response and paper revision. Some of the minor concerns were clarified. Unfortunately, I still believe that the technical novelty of the paper remain insufficient at the current form.

---

> ### Author Response · Authors · 2021-11-12
> **Response to the concern and questions. Part1.**
>
> >Weakness:
> >- I have a hard time to find anything that is technically novel. All the training losses are inherent from StartGANv2 or CycleGAN. The main difference lies in the foreground/background separation, but there was not ablation study to validate this.
>
> We apologize if the technical novelty did not become clear in the submission and would like to emphasize the novelties introduced in our work in the following:
>
> - We compute both style and content from a given input instead of only style as in StarGAN v2. It is crucial to separate them as shown in fig. 5 (a) and (b). In fig. 5 (a), the baseline StarGAN v2 cannot differ between style and content, which results in output images that use the background of the reference images (style/content column) instead of the source images. Moreover, without the style/content separation, the style computed by StarGANv2 doesn’t encode the structural patten of the reference images thus the generated defect patterns are simplistic and repetitive (e.g., there is no full scratch in the second row of fig 5(a)).
>
> - We enforce the fg/bg separation by simple but yet effective classification losses (the FG classifier and the BG classifier). Without the classification losses, the model was not constrained to encode the foreground contents into the designed channels at the information bottle neck of the generator. Please compare column (a) to (d) in fig. 5 and one can see that in all rows the correct foreground and background types are generated, but with a shift in brightness which will be addressed in point 3 later.
>
>     The advantage of fg/bg separation is three-fold: first, with the fg/bg separation the Mapping Network and Style-Content Encoder in our model are able to focus on learning the distribution of the foreground contents (e.g., defects). Second, this feature allows our model to collect foreground contents (defects) over different backgrounds (products) to reduce the need for data. In this way, we don’t need to have abundant data for a single product, instead, we can make use of little data from different products that share similar defect patterns. Third, it allows free combination of fg/bg which enriches the diversity of the outputs.
>
>     The models that were trained without fg/bg separation (e.g., StyleGAN v2, BigGAN and StarGAN v2) require more data of a single product for training. For example, StyleGAN v2 and BigGAN suffered from mode collapsing on the SDI dataset. Also, they showed signs of overfitting by generating images similar to the training data. Moreover, without the fg/bg separation, the foreground contents in the generated images are conditioned on the given backgrounds for all three methods. That is, a scratch pattern that present on product A will not be put to product B in the generated images which limits the diversity of the outputs.
>
> - We separately decoded the foreground and background in the generator. Note that the style is only applied to the foreground. With this modification, the model is forced to focus on learning the style of foreground contents (defects) instead of the style of the background. In fig. 5(e), we can see a clear effect after applying this feature: the backgrounds provided by the source images are better preserved than before (with the correct brightness).
>
> - Mokady et al. introduced a concept that the model should be able to identity the difference between ‘two’ domains when one of the domains contains a feature that the other doesn’t have. We refer to this concept as “anchor” and combine it with the multi-task discriminator to work with multiple domains (> 2).  Therefore, our model can not only perform the transformation between non-defective and defective samples, but also between defective samples (see fig. 10 in appendix E).
>
>     The quantitative evaluation of above-mentioned components can be found in Appendix E.
>
> The combination of all these ingredients leads to a model that cannot only transfer existing foreground contents (defects) across different backgrounds (products) but also apply new foreground contents generated from latent codes to the assigned backgrounds. Furthermore, our model can generate images with higher diversity and more realistic appearance when only limited data for each product is available, which StarGAN v2 and other state-of-the-art image synthesis methods could not do before. This is particularly useful when there are only few images with a specific combination of product and defect but plenty of similar defects on other products.
>
> In fig. 5(e), we already presented the ablation of fg/bg separation (the baseline StarGAN v2 does not have this feature). Would you have additional suggestions how to extend the ablation study?

---

> ### Author Response · Authors · 2021-11-22
> **Thank you for the advice, we revised the paper with regard to your suggestions.**
>
> >Weakness:
> >-   I have a hard time to find anything that is technically novel. All the training losses are inherent from StartGANv2 or CycleGAN. The main difference lies in the foreground/background separation, but there was not ablation study to validate this.
>
> We apologize if the technical novelty did not become clear in the first submission. We now revised Section 3 in the main paper and emphasized more on our novelties.
>
> >-   Only results on a _closed_ dataset is reported. I feel that this paper would be a lot more better suited for conferences with a focus on applications. I am not sure what’s the take-away lesson on the technical components from this paper.
>
> In the revised version, please find the experiment results on the MVTec AD Dataset in Appendix E.4. Also, we revised the Conclusion to make the take-away lesson clear.

---

### Official Review · Reviewer_GkV1 · 2021-11-07

**Correctness:** 3
**Technical Novelty And Significance:** 2
**Empirical Novelty And Significance:** 2
**Recommendation:** 3
**Confidence:** 4

**Main Review:**

Detailed comments are as follows.

1. The proposed method is reasonable, but the technical novelty is only moderate. The GAN formulation described in the paper heavily relies on StarGAN v2, with additional addition of the content transfer strategy introduced in Mokady et al., 2020.

2. The presentation needs significant improvements. For example, the contents/writing of Section 3 are way too similar to StarGAN v2. This could negatively affect the evaluation of the technical contributions of this work.

3. The experimental results are not convincing. The experiment is tested on only the SDI dataset, and the comparisons do not include existing SOTA techniques for defect classification. It is not clear whether the claimed main advantage of increasing the diversity of defects on each specific target product would still be valid for the case that defects of different products vary significantly. Also, it would be more insightful to include the experiment of defect localization (say, over the MVTec AD dataset) to further justify the usefulness of the proposed approach.

4. The ablation study on page 7 should be made more comprehensive (e.g., with explicit quantitative results), rather than with qualitative examples as in Figure 5.

5. The paragraph of cross-domain effect (with respect to Table 4) needs to be discussed in more detail. Is the defect transferring still effective for the case that defects of different products vary significantly?


**Summary Of The Paper:**

This paper describes a StarGAN-based approach to generating synthetic images of various defects based on the underlying training data. The proposed method considers two different types of domains, e.g., the foreground domain as defect types and the background domain as product types, and the derived synthetic images are included in the training data to improve the performance of defect classification.

**Summary Of The Review:**

The technical novelty of the proposed method can be improved, and the experiments lack comparisons with SOTA techniques for defect classification. This work is not ready for publication yet.

---

> ### Author Response · Authors · 2021-11-12
> **Response to the concerns and questions. Part 2.**
>
> >2.  The presentation needs significant improvements. For example, the contents/writing of Section 3 are way too similar to StarGAN v2. This could negatively affect the evaluation of the technical contributions of this work.
>
> Thank you for the kind advice, we will revise Section 3 and emphasize more on the difference between our work and the related works.
>
> >3.  The experimental results are not convincing. The experiment is tested on only the SDI dataset, and the comparisons do not include existing SOTA techniques for defect classification.
>
> It is not clear whether the claimed main advantage of increasing the diversity of defects on each specific target product would still be valid for the case that defects of different products vary significantly.
>
> Also, it would be more insightful to include the experiment of defect localization (say, over the MVTec AD dataset) to further justify the usefulness of the proposed approach.
>
> Additional results from experiments conducted on a subset of the MVTec AD dataset will be added to the supplementary material to show our model can adapt to more products than the three products in the SDI dataset. The subset was chosen because as you pointed out correctly, our model was not trained to tackle the localization explicitly. However, we believe it can be achieved by a slight modification in the current architecture and extra constrains (losses). We will discuss this limitation more clearly in the revised version of our paper and leave potential improvements to future work.
>
> Please note that the classification task is just one possible downstream task for our generative model. Still we would like to improve this part of the paper and we’re wondering if you have specific SOTA techniques for defect classification in mind?
>
> Our work builds on an assumption that the patterns of defects on the surface of different products resemble each other. At the moment, the case like transferring defects between product A with defect types (i, j, k) and product B with defect types (i, j, m) works theoretically and produces an image of product A with defect type m. However, training a classifier on this kind of data would only make sense if product A with defect m actually exists in the real-world. Does this address your concern?
>
> >3.  The ablation study on page 7 should be made more comprehensive (e.g., with explicit quantitative results), rather than with qualitative examples as in Fig. 5.
>
> Sorry for the information that was missing in the main paper. The quantitative results were already presented in the supplementary material (please see Appendix E). However, we forgot to point it out in the main paper. We will add the missing reference in the revision.
>
> >4.  The paragraph of cross-domain effect (with respect to Table 4) needs to be discussed in more detail. Is the defect transferring still effective for the case that defects of different products vary significantly?
>
> We discussed significantly different defects already above (at point 3). We will revise the paragraph that describes Table 4.
>
> Thank you for your comments!

---

> ### Author Response · Authors · 2021-11-12
> **Response to the concerns and questions. Part 1.**
>
> >1. The proposed method is reasonable, but the technical novelty is only moderate. The GAN formulation described in the paper heavily relies on StarGAN v2, with additional addition of the content transfer strategy introduced in Mokady et al., 2020.
>
> We apologize if the technical novelty did not become clear in the submission and would like to emphasize the novelties introduced in our work in the following:
>
> - We compute both style and content from a given input instead of only style as in StarGAN v2. It is crucial to separate them as shown in fig. 5 (a) and (b). In fig. 5 (a), the baseline StarGAN v2 cannot differ between style and content, which results in output images that use the background of the reference images (style/content column) instead of the source images. Moreover, without the style/content separation, the style computed by StarGANv2 doesn’t encode the structural patten of the reference images thus the generated defect patterns are simplistic and repetitive (e.g., there is no full scratch in the second row of fig 5(a)).
>
> - We enforce the fg/bg separation by simple but yet effective classification losses (the FG classifier and the BG classifier). Without the classification losses, the model was not constrained to encode the foreground contents into the designed channels at the information bottle neck of the generator. Please compare column (a) to (d) in fig. 5 and one can see that in all rows the correct foreground and background types are generated, but with a shift in brightness which will be addressed in point 3 later.
>
>     The advantage of fg/bg separation is three-fold: first, with the fg/bg separation the Mapping Network and Style-Content Encoder in our model are able to focus on learning the distribution of the foreground contents (e.g., defects). Second, this feature allows our model to collect foreground contents (defects) over different backgrounds (products) to reduce the need for data. In this way, we don’t need to have abundant data for a single product, instead, we can make use of little data from different products that share similar defect patterns. Third, it allows free combination of fg/bg which enriches the diversity of the outputs.
>
>     The models that were trained without fg/bg separation (e.g., StyleGAN v2, BigGAN and StarGAN v2) require more data of a single product for training. For example, StyleGAN v2 and BigGAN suffered from mode collapsing on the SDI dataset. Also, they showed signs of overfitting by generating images similar to the training data. Moreover, without the fg/bg separation, the foreground contents in the generated images are conditioned on the given backgrounds for all three methods. That is, a scratch pattern that present on product A will not be put to product B in the generated images which limits the diversity of the outputs.
>
> - We separately decoded the foreground and background in the generator. Note that the style is only applied to the foreground. With this modification, the model is forced to focus on learning the style of foreground contents (defects) instead of the style of the background. In fig. 5(e), we can see a clear effect after applying this feature: the backgrounds provided by the source images are better preserved than before (with the correct brightness).
>
> - Mokady et al. introduced a concept that the model should be able to identity the difference between ‘two’ domains when one of the domains contains a feature that the other doesn’t have. We refer to this concept as “anchor” and combine it with the multi-task discriminator to work with multiple domains (> 2).  Therefore, our model can not only perform the transformation between non-defective and defective samples, but also between defective samples (see fig. 10 in appendix E).
>
>     The quantitative evaluation of above-mentioned components can be found in Appendix E.
>
> The combination of all these ingredients leads to a model that cannot only transfer existing foreground contents (defects) across different backgrounds (products) but also apply new foreground contents generated from latent codes to the assigned backgrounds. Furthermore, our model can generate images with higher diversity and more realistic appearance when only limited data for each product is available, which StarGAN v2 and other state-of-the-art image synthesis methods could not do before. This is particularly useful when there are only few images with a specific combination of product and defect but plenty of similar defects on other products.

---

> ### Author Response · Authors · 2021-11-22
> **Thank you for the advice, we revised the paper with regard to your suggestions.**
>
>
>
> >1.  The proposed method is reasonable, but the technical novelty is only moderate. The GAN formulation described in the paper heavily relies on StarGAN v2, with additional addition of the content transfer strategy introduced in Mokady et al., 2020.
> >2.  The presentation needs significant improvements. For example, the contents/writing of Section 3 are way too similar to StarGAN v2. This could negatively affect the evaluation of the technical contributions of this work.
>
> Thank you for the kind advice, we apologize if the technical novelty did not become clear in the first submission and have revised the Section 3 accordingly.
>
> >3.  The experimental results are not convincing. The experiment is tested on only the SDI dataset, and the comparisons do not include existing SOTA techniques for defect classification.
> It is not clear whether the claimed main advantage of increasing the diversity of defects on each specific target product would still be valid for the case that defects of different products vary significantly.
> Also, it would be more insightful to include the experiment of defect localization (say, over the MVTec AD dataset) to further justify the usefulness of the proposed approach.
>
> In the revised version, please find the experiment results on the MVTec AD Dataset in Appendix E.4. There we also provide results that address your concern about the cross-domain effect.
>
> >4.  The ablation study on page 7 should be made more comprehensive (e.g., with explicit quantitative results), rather than with qualitative examples as in Fig. 5.
>
> We added the missing reference in the revision and please find the quantitative results in Appendix E.3.
>
> >5.  The paragraph of cross-domain effect (with respect to Table 4) needs to be discussed in more detail. Is the defect transferring still effective for the case that defects of different products vary significantly?
>
> We revised the paragraph that describes Table 4 and added the discussion about cross-domain effect in Appendix E.4.

---

> > ### Comment · Reviewer_GkV1 · 2021-11-30
> > **Concluding remarks & suggestions**
> >
> > I appreciate the authors’ efforts in addressing my concerns, and improving the quality of the presentation. However, the technical novelty of this work needs to be strengthened so that it can be a strong candidate for the upcoming top-tier conferences.

---

### Decision · Program_Chairs · 2022-01-20

**Decision:**

Reject

**Comment:**

The paper proposes a GAN based method for synthesizing various types of defects as foreground on different product images (background). The method builds upon StarGANv2, and adds the cycle/content consistency loss and classification loss between foreground and background. While the paper considers an important problem/application, the reviewers found it lacking sufficient novelty for publication. The paper will be more suited for publication at an application oriented venue.